# An improved harris hawks optimization algorithm based on chaotic sequence and opposite elite learning mechanism

**Ting Yang**[1], **Jie Fang**[1‡], **Chaochuan Jia** [2,3☯*], **Zhengyu Liu**[2,3‡], **Yu Liu**[2,3‡]

**1** College of Electronic and Optoelectronic Engineering, West Anhui University, Lu'an, China, **2** College of Electronics and Information Engineering, West Anhui University, Lu'an, China, **3** Intelligent networked vehicle laboratory, West Anhui University, Lu'an, China

☯ These authors contributed equally to this work.
‡ JF, ZL and YL also contributed equally to this work.
* ccjia@hfcas.ac.cn

**Data Availability Statement:** All relevant data are within the paper.

**Funding:** This work was partially supported by Natural Science research project of Universities in Anhui Province(NO.KJ2021A0953), Natural

## Abstract

The Harris hawks optimization (HHO) algorithm is a new swarm-based natural heuristic algorithm that has previously shown excellent performance. However, HHO still has some shortcomings, which are premature convergence and falling into local optima due to an imbalance of the exploration and exploitation capabilities. To overcome these shortcomings, a new HHO variant algorithm based on a chaotic sequence and an opposite elite learning mechanism (HHO-CS-OELM) is proposed in this paper. The chaotic sequence can improve the global search ability of the HHO algorithm due to enhancing the diversity of the population, and the opposite elite learning can enhance the local search ability of the HHO algorithm by maintaining the optimal individual. Meanwhile, it also overcomes the shortcoming that the exploration cannot be carried out at the late iteration in the HHO algorithm and balances the exploration and exploitation capabilities of the HHO algorithm. The performance of the HHO-CS-OELM algorithm is verified by comparison with 14 optimization algorithms on 23 benchmark functions and an engineering problem. Experimental results show that the HHO-CS-OELM algorithm performs better than the state-of-the-art swarm intelligence optimization algorithms.

## 1. Introduction

The optimization issues in real-world problems have received increasing attention from researchers in the fields of artificial intelligence [1], computer vision [2], compressed sensing [3, 4], decision-making [5] and engineering for practical applications [6]. Traditional algorithms are based on derivative methods due to their mathematical complexity, which can only be used to deal with small-scale problems that must be continuous and derivable [7]. Therefore, it is difficult to achieve global optimization for multimodal functions and dynamically changing, strongly nonlinear problems using traditional algorithms. To solve complex and large-scale problems, many swarm intelligence (SI) optimization algorithms that imitate

Science Key Scientific Research Project of West Anhui University (NO. 0041021003, WXZR201903).

**Competing interests:** The authors have declared that no competing interests exist.

swarm behaviour in natural phenomena, including Cuckoo Search (CS) [8], Grey Wolf Optimizer (GWO) [9], Particle Swarm Optimization (PSO) [10], Artificial Bee Colony (ABC) [11], Suffled Frog Leaping Algorithm (SFLA) [12], Whale Optimization algorithm (WOA) [13], Gravitational Search algorithm (GSA) [14], Jaya [15] and Harris Hawk Optimization (HHO) [16] have been proposed. All SI algorithms have two search phases: global exploration, which searches the whole space for a promising area, and local exploitation, which searches a chosen area that is promising to contain the best solution. However, a single SI algorithm can not deal with all optimization problems. Still, the algorithms proposed recently or those that are not yet discovered have a wide range of application prospects.

HHO is a new swarm intelligence optimization algorithm proposed by Heidari et al. [16] in 2019 that mimics the way Harris eagles find and chase prey in nature, including global exploration, local besiege and pounce behaviour. HHO has been widely applied to address the optimization of functions and engineering applications due to its gradient-free and powerful nature with high performance. Heidari et al. used HHO to optimize 29 benchmark functions and 6 engineering applications, and the results show that HHO has better competitiveness and application prospects than other SI algorithms [16]. Houssein et al. [17] used the HHO in combination with the k-nearest neighbours and the support vector machines for chemical compound activities and descriptor selection, respectively. To denoise the satellite images, Golilarz et al. [18] determined optimal wavelet coefficients by using the HHO. HHO was applied to optimize the water network distribution of Homashahr city in Iran in [19]. Abbasi et al. [20] utilized HHO to microchannel heat sinks to minimize entropy generation. Jiao et al. [21] and Liu et al. [22] used HHO to find the optimal parameters of photovoltaic models. However, similar to the other SI algorithms, the HHO still has some limitations, such as the multiplicity of solutions generated by a randomized policy that is finite in the initialization phase. Moreover, because global exploration is only performed in the first half of the iteration, it is difficult to balance the global exploration and local exploitation capacities by using the escaping energy of prey, so the algorithm may converge slowly, has low solution accuracy and prematurely falls into a local optimal solution.

To conquer the limitations of HHO, many HHO variant algorithms have been proposed. For example, Ali et al. [23] used the best solution to deal with the boundary condition instead of the boundary of the search space in HHO. Hu et al. [24] also proposed an improved HHO algorithm, which embedded the velocity into the exploration phase and updated the solutions by using the crossover operator of the artificial tree algorithm in the exploitation phase. A boosted HHO (BHHO) technique was proposed by Houssein et al. [25]. BHHO used the mutation of the DE algorithm and the flower pollination process of flowering plants instead of the hard and soft siege with progressive rapid dives, respectively. Afterward, Houssein et al. [26] proposed a hybrid HHO algorithm (HHHO), which used a chaos map to update the escaping energy function to balance the exploration and exploitation phases. In addition, a cuckoo search algorithm was used to update the optimal and random solutions to improve the global search capability. Mohamed et al. [27] proposed an improved HHO algorithm, which employed the salp swarm algorithm to balance the exploitation and exploration capabilities. Qu et al. [28] also put forward an HHO variant algorithm, which was based on the information exchange between Harris's hawks, while chaos disturbance was used to update the escaping energy function to balance the local and global search capabilities. Shi et al. [29] applied the grey wolf optimizer and the salp swarm algorithm to improve the search capability of HHO. Aneesh et al. [30] introduced mutation interval to update the escaping energy function and used the average fitness to determine the updating strategies of Harris hawks in the exploration phase. Chen et al. [31] came up with a new framework of HHO, which combined the chaos map, multi population and differential evolution mechanism to improve the performance of

HHO. Ahmed et al. [32] proposed a chaotic Harris's hawk optimization (CHHO) algorithm in which the chaotic sequences that are generated by the ten chaotic equations are used instead of the random parameter $q$ in the exploration phase. Singh [33] also used chaotic sequences instead of random parameters, which are $r1$ and $r2$ in the exploration phase and vector $S$ in the exploitation phase (CSHHO). Chen et al. [34] proposed a diversification-enriched Harris hawk optimization (DEHHO), which embedded the chaotic sequence to search the neighbourhood of the current optimal solution and introduced the OBL mechanism to enlarge the search area in the whole space.

Although the improvement strategies mentioned above have enhanced the capability of the standard HHO algorithm to a certain extent, it can still be improved by other strategies. Inspired by [32–34], an improved HHO variant algorithm is proposed in this paper. The contributions of this paper are as follows: (i) a chaotic sequence chaotic sequence recombination mechanism (CSRM) strategy is proposed, which enhances the distribution of the initialized solutions in the search space, and accelerates the convergence rate of HHO;(ii) the generalized opposition-based learning recombination mechanism (OBLRM) is proposed, which can have the opportunity to carry out global search in the later period of iteration to jump out of the local optimum and improve the accuracy of the solution. The rest of the paper is organized as follows. Section 2 gives a detailed overview of the HHO algorithm. The proposed method is introduced in detail in Section 3. In Section 4, the experimental results are analysed. Finally, the conclusions are presented in Section 5.

## 2. Background studies

HHO is a new swarm intelligence optimization algorithm proposed by Heidari et al. [16] in 2019 that mimics the way Harris eagles find and chase prey in nature, which has strong global search capability and adjusts few parameters. The whole foraging process mainly consists of three phases: the exploration phase, transition from the exploration phase to the exploitation phase and the exploitation phase. All phases of HHO are shown in (Fig 1), and each phase is presented in detail as follows.

### 2.1 Exploration phase

During the exploration phase, Harris hawks primarily search for prey, which may be a rabbit. When Harris hawks detect and track a rabbit with their keen eyes, two strategies are used to update their locations, which can be formulated as:

$$X_i(t+1) = \begin{cases} X_{rand}(t) - r_1|X_{rand}(t) - 2r_2X_i(t)|, & q \geq 0.5 \\ (X_{rabbit}(t) - X_m(t)) - r_3(LB + r_4(UB - LB)), & q < 0.5 \end{cases} \quad (1)$$

where $X_i(t)$ and $X_{rabbit}(t)$ are the positions of the $i^{th}$ hawk and the rabbit, respectively, at the current iteration, t.$X_{rand}(t)$ is the position of the randomly selected hawk, $X_i(t+1)$ is the position of the $i^{th}$ hawk at the next $(t+1)^{th}$ iteration, $r_1$, $r_2$, $r_3$ and $r_4$ are four random numbers between [0, 1], $q$ is a random number between [0,1] that is applied to switch the strategy, and $X_m(t)$ is the mean position of the current population as follows:

$$X_m(t) = \frac{1}{N_p}\sum_{i=1}^{N_p} X_i(t) \quad (2)$$

where $N_p$ is the size of the population, and [$LB$, $UB$] denotes the search space.

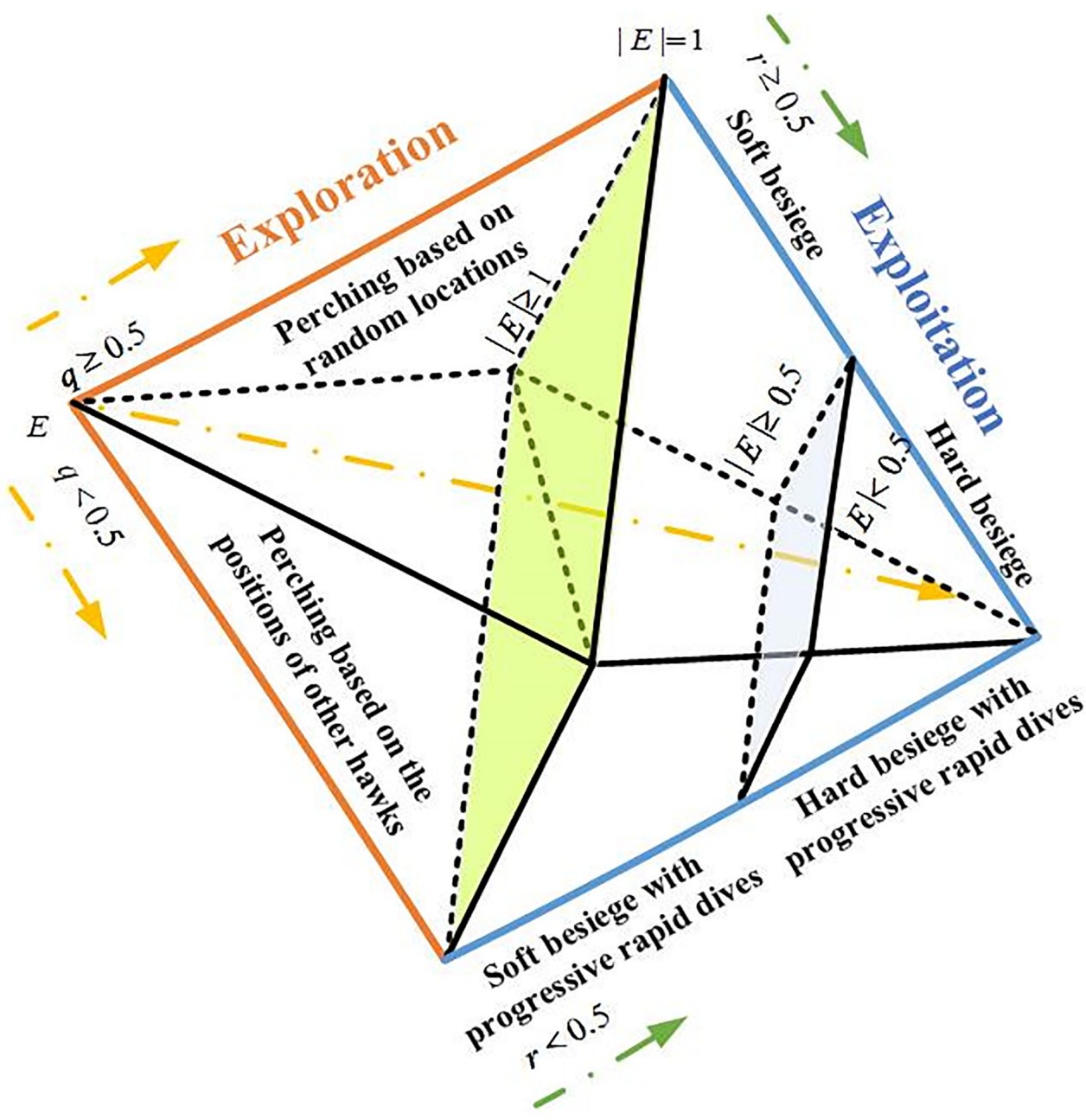

**Fig 1. Different phases of HHO (Heidari et al.[16]).**

## 2.2 Transition from exploration to exploitation

The transition from the global search (exploration) to local search (exploitation) of Harris hawks mainly depends on the escaping energy $E$ of the prey (rabbit), where $E$ can be calculated as follows.

$$E = 2E_0(1 - \frac{t}{T}) \tag{3}$$

where $E_0$ is the random between (-1,1) in each iteration and $T$ denotes the maximum number

of iterations. Thus, the escaping energy $E$ is within the interval (-2, 2). When $|E| \geq 1$, it indicates that the rabbit is capable of escaping, so the Harris hawks perform a global search (exploration). When $|E| < 1$, it indicates that the rabbit is weak, and the Harris hawks perform a local search (exploitation).

### 2.3 Exploitation phase

When a rabbit is spotted, Harris hawks will besiege to the rabbit and wait for the chance to pounce. However, the rabbit may escape the encirclement during besieging, so Harris hawks should constantly adjust their flight strategies according to the behaviour of the rabbit. Four strategies, which will be switched by the escaping energy $E$ and a random $r$, will be used in the exploitation phase to mimic the hunting behaviour of a Harris hawk. Each strategy is introduced in detail as follows.

**2.3.1. Soft besiege.** When $|E| \geq 0.5$ and $r \geq 0.5$, the rabbit has enough energy to try to escape the siege by jumping at will, but is ultimately unable to escape, so Harris hawks can capture the rabbit by surrounding the rabbit and performing a surprise pounce. This strategy can be formulated as follows:

$$X(t+1) = \Delta X(t) - E \left| J X_{rabbit}(t) - X(t) \right| \tag{4}$$

$$\Delta X(t) = X_{rabbit}(t) - X(t) \tag{5}$$

$$J = 2(1 - r_5) \tag{6}$$

where $\Delta X(t)$ represents the difference between the optimal individual and the current individual, $J$ is the random jump strength of the rabbit and $r_5$ is a random number between (0,1).

**2.3.2. Hard besiege.** When $|E| < 0.5$ and $r \geq 0.5$, the rabbit is exhausted and has neither the energy nor the opportunity to escape, so the Harris hawks can capture the rabbit by surrounding the rabbit and performing a surprise pounce. This strategy can be formulated as follows:

$$X(t+1) = X_{rabbit}(t) - E \left| \Delta X(t) \right| \tag{7}$$

**2.3.3. Soft besiege with progressive rapid dives.** When $|E| \geq 0.5$ and $r < 0.5$, the rabbit has enough energy to successfully escape from its encirclement, so the Harris hawks need a more intelligent encirclement to surround the rabbit before performing a surprise pounce. Harris hawks surround the rabbit by performing the following two strategies; when the first strategy fails, the second strategy is performed.

$$\text{The first strategy is } Y = X_{rabbit}(t) - E \left| J X_{rabbit}(t) - X(t) \right| \tag{8}$$

$$\text{The second strategy is } Z = Y + S \times LF(D) \tag{9}$$

where $S$ is a random vector with $1 \times D$ dimensions, $D$ denotes the dimension of the search space, and $LF(\cdot)$ is the *Levy flight* function as follows:

$$LF(x) = 0.01 \times \frac{\mu \times \sigma}{|v|^{\frac{1}{\beta}}}, \sigma = \left( \frac{\Gamma(1+\beta) \times \sin\left(\frac{\pi\beta}{2}\right)}{\Gamma\left(\frac{1+\beta}{2}\right) \times \beta \times 2^{\left(\frac{\beta-1}{2}\right)}} \right)^{\frac{1}{\beta}} \tag{10}$$

where $\beta$ is a constant to be set to 1.5, and $\mu$ and $v$ are random numbers between (0, 1). Thus,

the updating strategy for this phase can ultimately be modelled as follows:

$$X(t+1) = \begin{cases} Y, & F(Y) < F(X(t)) \\ Z, & F(Z) < F(X(t)) \end{cases} \tag{11}$$

**2.3.4. Hard besiege with progressive rapid dives.** When $|E|<0.5$ and $r<0.5$, the rabbit may make a successful escape; however, its escape energy is insufficient, so the Harris hawks form a hard encirclement to surround the rabbit before performing a surprise pounce. They still perform two strategies to update their positions in this phase.

$$\text{The first strategy is } Y = X_{rabbit}(t) - E\,|\,JX_{rabbit}(t) - X_m(t)| \tag{12}$$

$$\text{The second strategy is } Z = Y + S \times LF(D) \tag{13}$$

Thus, the updating strategy for this phase can ultimately be formulated as follows:

$$X(t+1) = \begin{cases} Y, & F(Y) < F(X(t)) \\ Z, & F(Z) < F(X(t)) \end{cases} \tag{14}$$

In summary, (Fig 2) shows the optimization process of the basic HHO algorithm.

# 3. Proposed scheme

To improve the diversity of initial population and enhance the ability to jump out of local optimal solution of HHO algorithm, the specific implementation of the proposed algorithm is described in detail in this section. Two recombination mechanisms, the chaotic sequence recombination mechanism (CSRM) and generalized opposition-based learning recombination mechanism (OBLRM), are introduced to enhance the performance of the HHO algorithm. The improved HHO algorithm does not change the structure of the HHO. (Fig 3) shows the optimization process of the proposed algorithm.

## 3.1 Chaotic sequence recombination mechanism

Since the chaotic system can vary randomly, if the running time is unlimited, every state will be realised. This means that the chaotic maps can be applied to build the search basis of optimization methods, or introduced into some raw optimization algorithms to improve their exploration competence [35, 36]. Due to sensitivities of the initial condition, randomness and ergodicity of a chaotic sequence, it is often used in optimization algorithms to decrease the chance of premature maturation [37, 38]. So a chaotic sequence can significantly enhance the capability of the HHO algorithm by replacing the random values, as confirmed in the literature [26, 32–34]. Therefore, the chaotic sequence generated by logistic mapping is applied to generate the initial solutions in HHO. The logistic mapping can be modelled as follows:

$$u_{i+1} = c \cdot u_i(1 - u_i), \ i = 1, 2, \ldots .k; \ u_i \in (0, 1), \\ u_i \neq 0.25, 0.5 \text{ and } 0.75 \tag{15}$$

where $u_i$ represents the chaotic variable in the $i^{th}$ iteration, $k$ represents the iteration number, and $c$ is the control parameter, which is set to 4. At the initial phase of HHO, the chaotic search around the initial candidate solutions can enhance the diversity of the population and then improve the exploration capability of the algorithm. The initial population $P$ is generated

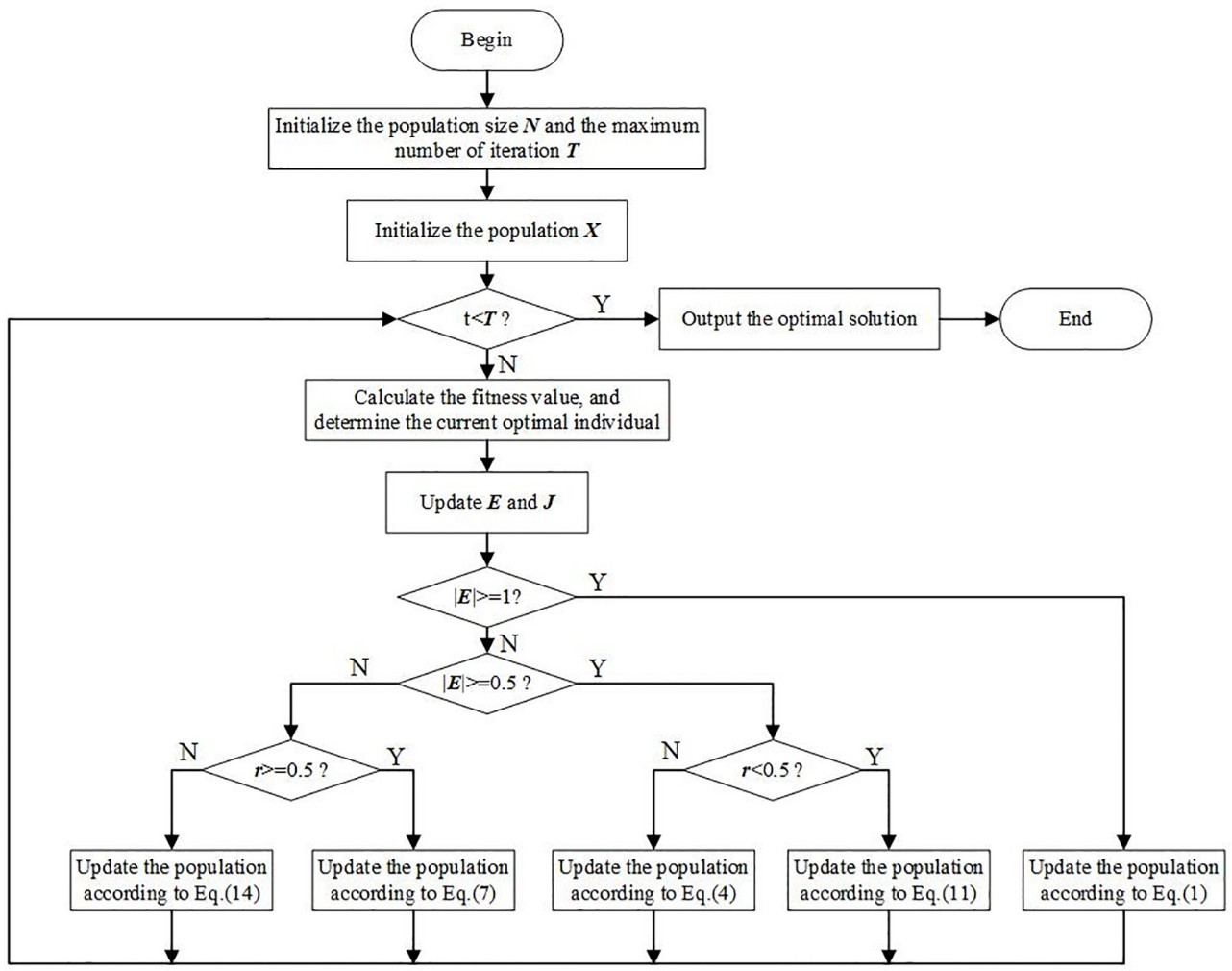

**Fig 2. Optimization process of basic HHO.**

according to the following equation:

$$x_i = LB + rand() * (UB - LB), \ i = 1, 2, \ldots . N_p \tag{16}$$

where $x_i$ denotes the $i^{th}$ candidate solution and $rand()$ is a function that generates a random number between [0, 1]. Then, an updated population $P_c$ is obtained by combining $P$ with the chaotic sequence $u_i$.

$$P_c = P + u_i \cdot P \tag{17}$$

The recombination mechanism is performed by recombining $P_c$ and $P$; then, a new population will be generated by selecting the solutions corresponding to the first $N_P$ fitness values. These steps need to be executed $k$ times, and finally a new initial population is obtained. Apart from the random distribution used by the standard HHO, the CSRM strategy enhances the distribution of the initialized solutions in the search space, thus speeding up the convergence of the HHO algorithm.

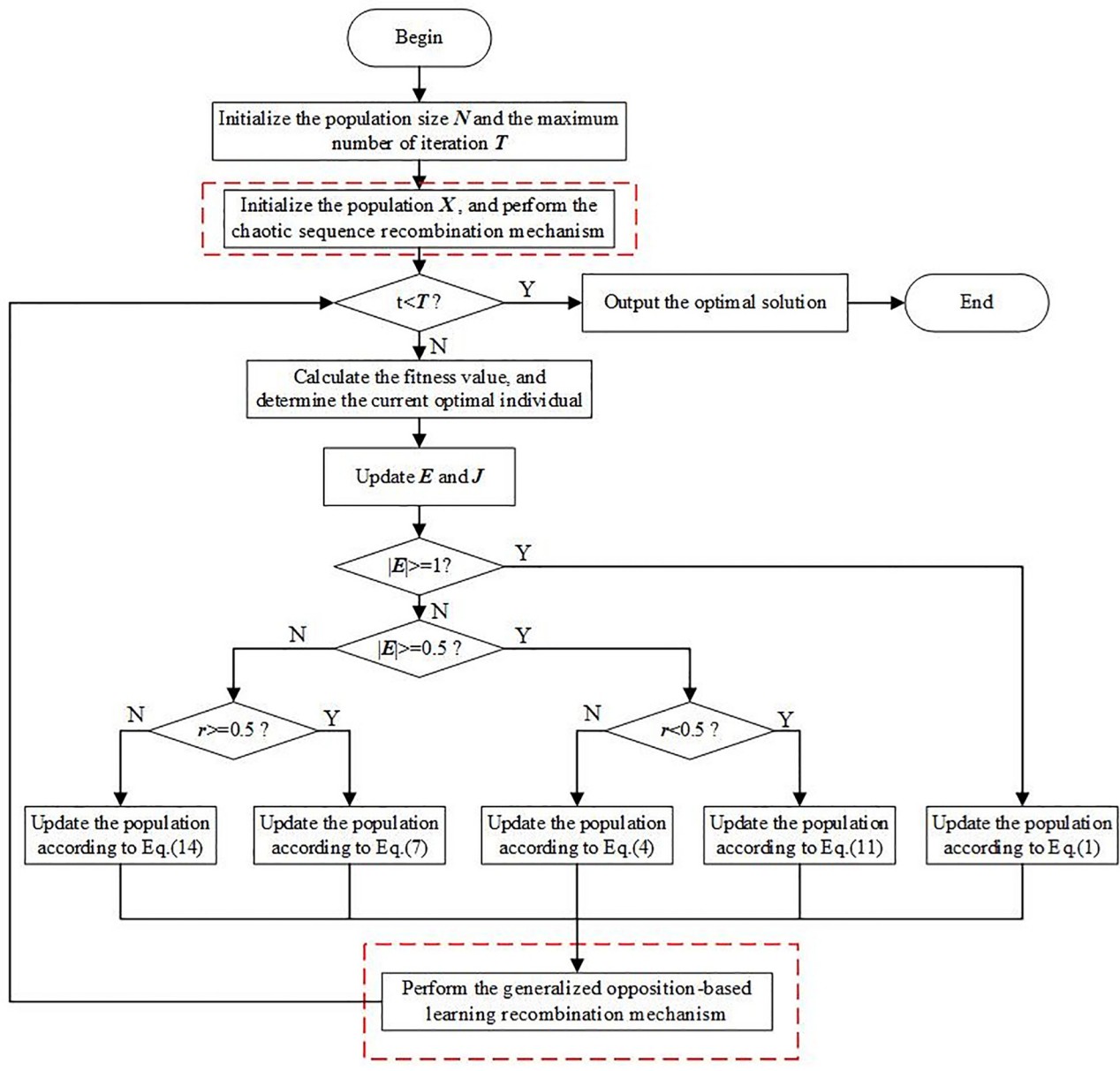

**Fig 3. Flow chart of the proposed HHO.**

## 3.2 Opposite elite learning recombination mechanism

The opposition-based learning (OBL) technique, which was first proposed in 2005 by Tizhoosh, is a machine intelligence strategy that aims to improve the capabilities of SI algorithms. Its core idea is to find a better solution between the current individual and the corresponding opposite solution, according to their fitness values. It has been verified that the OBL strategy can have more chances to approach the global optimal solution of the objective function [39]. Therefore, the OBL strategy has been widely applied by researchers to enhance the capabilities of SI algorithms, such as the WOA [40], GOA [41], PSO [42], SSA [43] and CS [44] algorithms.

Suppose that $x_i$ is the current individual; then the corresponding generalized opposite solution $\bar{x}_i$ can be calculated as follows:

$$\bar{x}_i = rand() * (UB + LB) - x_i, i = 1, 2, \ldots N_p \tag{18}$$

Then, a population $P$ composed of $x_i(i = 1,2,\ldots N_P)$ is the parent generation, while a population $P_o$ composed of $\bar{x}_i(i = 1, 2, \ldots N_p)$ is the offspring. Finally, the recombination mechanism is performed by recombining $P_o$ and $P$; therefore, a new population will be obtained by selecting the solutions corresponding to the first $N_P$ fitness values.

It is well known that in the HHO algorithm, even if the selected region is not globally optimal, the global search is no longer carried out in the later periods of iteration, so the HHO tends to converge to the local optimum prematurely. However, when the above generalized opposition-based recombination mechanism is embedded in the HHO algorithm, the improved algorithm can have the opportunity to carry out global search in the later period of iteration to jump out of the local optimum and improve the accuracy of the solution.

## 4. Experimental results and analysis

To evaluate the performance of the HHO-CS-OELM algorithm, two experiments are implemented in this section. In the first experiment, the proposed HHO-CS-OELM algorithm was compared with 14 SI algorithms, such as PSO, SFLA, ABC, WOA, CS, GSA, GWO, Jaya, BHHO, HHHO, CHHO, CSHHO, DEHHO and HHO, that were applied to optimize the 23 benchmark functions [13]. Second, the HHO-CS-OELM algorithm was applied to optimize the thresholds and weights of the back propagation (BP) neural network for UWB indoor positioning. All experiments are carried out on a Windows 10 operating system with MATLAB R2019a on a PC with Inter(R) core i7-10750H and 16 GB RAM memory.

### 4.1 Benchmark functions experiment

The HHO-CS-OELM algorithm and the other 14 SI algorithms are applied to optimize the 23 benchmark functions, which are categorised as unimodal, multimodal and fixed dimension multimodal. F1-F7 are unimodal functions for which there is only one global optimal solution. These functions can be utilized to evaluate the convergence speed and exploitation capability of the proposed algorithm. On the other hand, F8-F13 and F14-F23 are multimodal and fixed dimension multimodal functions, respectively, which have one global optimal solution and several local optimal solutions. These functions can be applied to evaluate the local optimal avoidance and exploration capabilities of the proposed algorithm. The details of these benchmark functions are provided in the literature [13]. Typical two-dimensional diagrams of some of these functions are shown in (Fig 4), from which, the prominent characteristics of these functions can be observed, Fig 4(a) and 4(b) are unimodal functions which have only one minimum value, however, from Fig 4(c) to 4(f) are multimodal functions which have a lot of local minimum values.

For all experiments, we set the population size to 30, the maximum number of iterations to 500, and the parameters of each algorithm are taken from the literature. Each algorithm is executed independently for 51 times for each function. The average and standard deviation results of these benchmark functions are recorded in Table 1, and the convergence curves of F1-F7, F8-F13 and F14-F23 are shown in Figs 5–7, respectively.

**4.1.1. Evaluation of exploitation capability (F1-F7).** The unimodal functions can be utilized to evaluate the exploitation capabilities of the SI algorithms because they only have one global optimal solution. The results in Table 1 show that the HHO-CS-OELM algorithm is highly competitive compared to other HHO variants and SI algorithms. For all unimodal

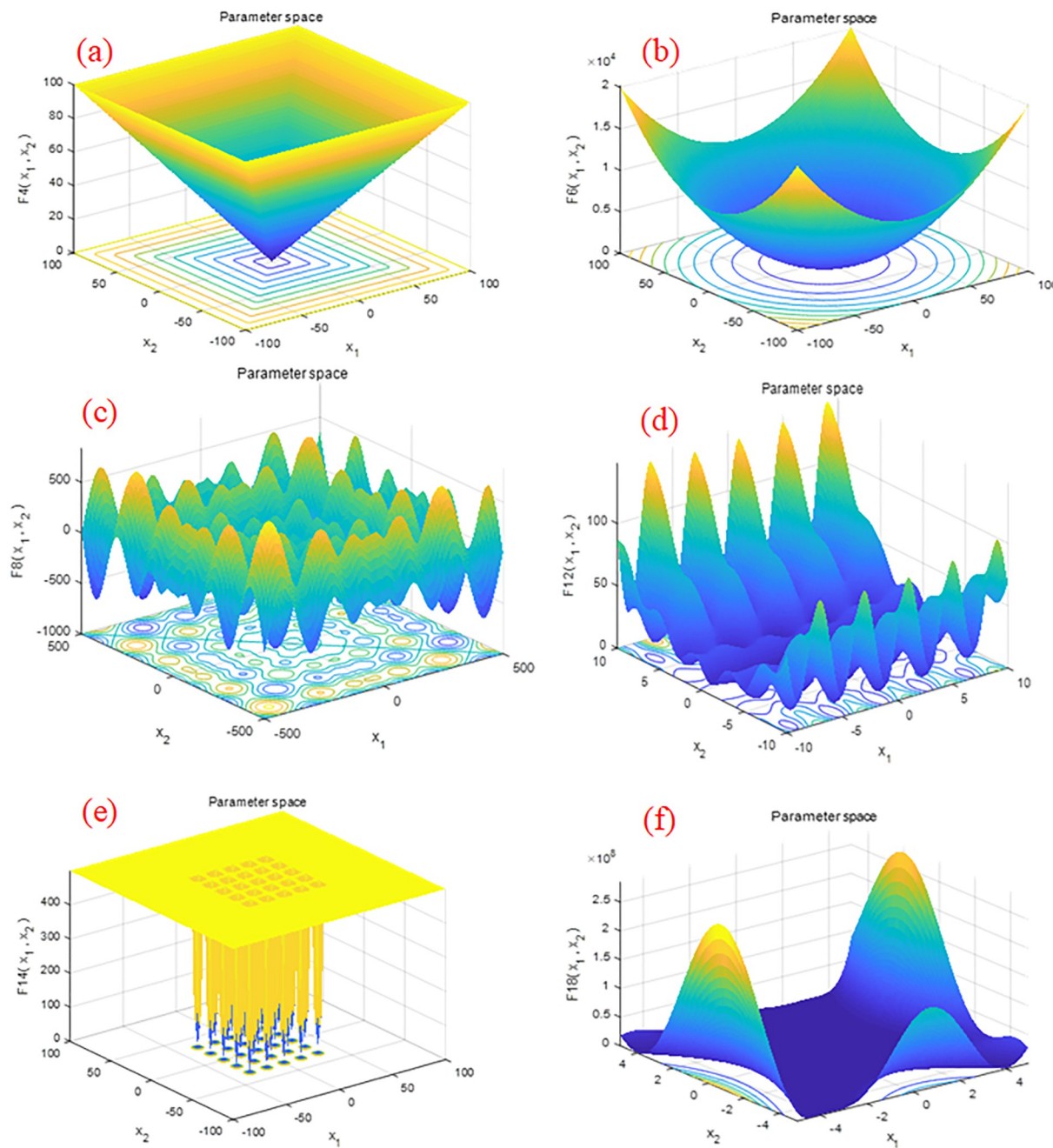

**Fig 4.** Typical 2D representations of benchmark mathematical functions: (a), (b) unimodal functions, (c), (d) multimodal functions, and (e), (f) fixed-dimension multimodal functions.

functions excluding F6, the HHO-CS-OELM algorithm acquires the best optimal average values and standard deviations, which indicates that the accuracy and stability of the HHO-CS-OELM algorithm are the best among all algorithms. On the other hand, the results in Fig 5 show that the convergence of the HHO-CS-OELM algorithm is the fastest. In summary, the exploitation capability of the HHO-CS-OELM algorithm is more competitive than that of other SI optimization algorithms.

**Table 1. Optimization results of 23 benchmark functions.**

| F | GSA [14] | | GWO [9] | | Jaya [15] | | PSO [10] | | SFLA [12] | |
|---|---|---|---|---|---|---|---|---|---|---|
| | ave | std | ave | std | ave | std | ave | std | ave | std |
| F1 | 2.41E-16 | 1.62E-16 | 5.71E-28 | 4.37E-28 | 0.0140947 | 0.0071817 | 1.848079 | 0.961603 | 5.74E-20 | 8.45E-20 |
| F2 | 0.0404513 | 0.0646169 | 7.38E-17 | 5.95E-17 | 1.0211176 | 0.4472688 | 0.293606 | 0.144564 | 1.58E-11 | 8.93E-12 |
| F3 | 739.13499 | 232.35362 | 2.02E-06 | 2.73E-06 | 40372.169 | 5855.3266 | 9590.704 | 687.5929 | 0.90268272 | 1.0032263 |
| F4 | 6.2525518 | 1.9013357 | 7.34E-07 | 3.73E-07 | 69.965142 | 6.9024873 | 20.81261 | 3.473756 | 0.03860862 | 0.0333454 |
| F5 | 71.111986 | 54.204282 | 27.06274 | 0.5907819 | 127.29214 | 100.5347 | 1253.756 | 938.8401 | 20.7200987 | 0.8355175 |
| F6 | 4.78E-16 | 5.12E-16 | 0.843911 | 0.701659 | 6.6326246 | 2.4593122 | 2.560659 | 2.148027 | **4.75E-19** | 4.89E-19 |
| F7 | 0.0632492 | 0.0214553 | 0.001221 | 0.0010094 | 7.8703457 | 2.2132186 | 0.127265 | 0.035116 | 0.00766484 | 0.0037064 |
| F8 | -2701.578 | 204.01577 | -5778.11 | 1589.8457 | -6260.097 | 394.13716 | -10601.6 | 340.3485 | -7853.7928 | 1146.5375 |
| F9 | 128.5475 | 55.882554 | 2.432101 | 2.1247137 | 377.81028 | 76.583244 | 221.9685 | 36.91306 | 72.2338468 | 14.717256 |
| F10 | 1.12E-08 | 2.33E-09 | 1.02E-13 | 1.52E-14 | 16.035105 | 1.5023415 | 1.960284 | 0.41207 | 0.2310297 | 0.5165981 |
| F11 | 25.101091 | 7.0435253 | 0.006549 | 0.0146435 | 0.3032846 | 0.1496428 | 1.029943 | 0.015273 | 0.0068969 | 0.0068333 |
| F12 | 1.9981369 | 0.9194609 | 0.03531 | 0.0070515 | 5.1036967 | 1.495743 | 5.789422 | 2.106513 | **2.4712E-15** | 8.89E-15 |
| F13 | 7.7274859 | 6.141158 | 0.6514 | 0.2613722 | **1.55E-32** | 4.41E-33 | 25.76373 | 4.462665 | 1.61E-19 | 1.12E-19 |
| F14 | 8.9614498 | 5.5141912 | 1.791644 | 1.086737 | 0.9980038 | 2.58E-09 | 0.998004 | 0 | 0.99800384 | 0 |
| F15 | 0.0047645 | 0.0019368 | 0.011581 | 0.0251343 | 0.0016676 | 0.0003386 | 0.008589 | 0.010989 | 0.00051041 | 0.0004537 |
| F16 | -1.031628 | 1.11E-16 | -1.03163 | 4.11E-08 | -1.031628 | 2.39E-07 | -1.03163 | 1.57E-16 | **-1.031628** | 0 |
| F17 | 0.3978874 | 0 | 0.397891 | 4.26E-06 | 0.3979032 | 2.11E-05 | 0.397887 | 0 | **0.39786736** | 0 |
| F18 | **3** | 5.56E-15 | 3.000031 | 2.65E-05 | 3.0001094 | 0.0001504 | **3** | 1.20E-15 | **3** | 5.87E-16 |
| F19 | -3.862782 | 3.85E-16 | -3.8627 | 7.94E-05 | -3.862782 | 0 | -3.86278 | 0 | -3.8627821 | 0 |
| F20 | -3.321995 | 3.14E-16 | -3.21049 | 0.0678094 | -3.249945 | 0.0641553 | -3.27444 | 0.06512 | -3.2506593 | 0.0651204 |
| F21 | -6.459084 | 3.7502678 | -8.13032 | 2.7674398 | -7.131524 | 4.1089097 | -5.10491 | 3.071476 | -6.1545785 | 3.7813392 |
| F22 | -10.40294 | 1.78E-15 | -10.4012 | 0.0010495 | -9.159937 | 2.3024201 | -8.87553 | 3.41539 | -10.402941 | 1.54E-15 |
| F23 | -9.417568 | 2.501806 | -10.5342 | 0.0008565 | -9.362327 | 2.4840699 | -7.29165 | 2.962045 | -8.1240606 | 3.3370575 |
| F | WOA [13] | | ABC [11] | | CS [8] | | BHHO [25] | | HHHO [26] | |
| | ave | std | ave | std | ave | std | ave | std | ave | std |
| F1 | 5.03E-73 | 6.93E-73 | 116.90328 | 99.446207 | 8.938738 | 4.0552844 | 4.30E-99 | 8.75E-99 | 4.06E-10 | 5.70E-10 |
| F2 | 1.70E-51 | 2.57E-51 | 66.274234 | 18.107702 | 14.18175 | 5.6171115 | 2.62E-51 | 4.01E-51 | 4.03E-07 | 7.27E-07 |
| F3 | 42368.289 | 11292.92 | 70874.315 | 13876.575 | 1933.585 | 396.47193 | 2.24E-67 | 5.00E-67 | 1.44E-06 | 3.22E-06 |
| F4 | 55.840991 | 31.524364 | 62.962487 | 2.9479726 | 10.72391 | 2.2298824 | 3.66E-49 | 8.00E-49 | 5.38E-06 | 1.18E-05 |
| F5 | 27.941996 | 0.4754027 | 3054695.1 | 1092768.8 | 728.4834 | 763.84808 | 0.3854 | 0.124802 | 0.0219886 | 0.035827 |
| F6 | 0.4217644 | 0.1741637 | 79.436661 | 49.36349 | 10.77344 | 2.87247 | 0.0004 | 0.0005828 | 0.0023816 | 0.0042346 |
| F7 | 0.0018998 | 0.00198 | 1.2834273 | 0.3349566 | 0.099935 | 0.0313949 | 0.0001634 | 0.0001834 | 0.0009076 | 0.0009345 |
| F8 | -10337.23 | 1840.1341 | **-4.58E+30** | 4.47E-30 | -7862.46 | 127.91114 | -12569.22 | 0.2997935 | -12568.58 | 1.8962802 |
| F9 | 0 | 0 | 430.15851 | 37.094244 | 286.506 | 57.40423 | **0** | **0** | 6.09E-08 | 1.36E-07 |
| F10 | 5.15E-15 | 2.97E-15 | 7.7570486 | 0.6390946 | 7.398407 | 3.4135373 | **8.88E-16** | **0** | 3.39E-06 | 6.71E-06 |
| F11 | 0.0349904 | 0.078241 | 1.9955217 | 0.6577877 | 1.06669 | 0.0253085 | **0** | **0** | 8.31E-10 | 1.69E-09 |
| F12 | 0.0191564 | 0.0055755 | 4594127.6 | 2431688.7 | 3.403843 | 0.7371567 | 9.99E-06 | 1.25E-05 | 2.69E-05 | 2.43E-05 |
| F13 | 0.4532223 | 0.1770201 | 16044917 | 7095440.6 | 7.367658 | 2.6789956 | 9.94E-05 | 0.0001138 | 0.0001264 | 0.0001767 |
| F14 | 3.3478595 | 4.233386 | 0.9980218 | 3.01E-05 | 0.998004 | 1.11E-16 | 1.1968093 | 0.4445424 | 0.9980039 | 4.54E-08 |
| f15 | 0.0005188 | 0.0001956 | 0.0010764 | 0.0001658 | 0.000424 | 0.0001288 | **0.0003011** | **3.18E-06** | 0.0008582 | 0.0008098 |
| F16 | -1.031628 | 7.49E-09 | -1.031628 | 9.98E-08 | -1.03163 | 1.11E-16 | -1.031628 | 2.79E-10 | -1.031623 | 7.80E-06 |
| F17 | 0.3978949 | 6.48E-06 | 0.3979008 | 1.41E-05 | 0.397887 | 4.40E-14 | 0.3978879 | 8.23E-07 | 0.3987115 | 0.0011985 |
| F18 | 3.0000377 | 6.56E-05 | 3.0000047 | 4.13E-06 | 3 | 1.37E-15 | 3.000001 | 1.53E-06 | 3.0000053 | 9.98E-06 |
| F19 | -3.854978 | 0.0076613 | -3.852782 | 2.66E-10 | -3.85278 | 6.28E-16 | -3.852755 | 2.42E-05 | -3.832871 | 0.0653854 |
| F20 | -3.211818 | 0.1029667 | -3.321931 | 0.0001428 | -3.322 | 3.45E-08 | -3.291531 | 0.0675993 | -3.182015 | 0.1164403 |
| F21 | -9.125273 | 2.2754774 | **-10.15320** | 8.85E-08 | -10.0532 | 9.13E-08 | -5.05512 | 6.57E-05 | -10.05253 | 0.0953224 |

*(Continued)*

**Table 1.** (Continued)

| F | CHHO [32] | | CSHHO [33] | | DEHHO [34] | | HHO [16] | | Proposed | |
|---|---|---|---|---|---|---|---|---|---|---|
| | ave | std | ave | std | ave | std | ave | std | ave | std |
| F22 | -6.148296 | 2.3727622 | **-10.40284** | 6.13E-08 | -10.3029 | 7.97E-07 | -4.623228 | 1.0383363 | -10.37496 | 0.0328237 |
| F23 | -6.730835 | 3.6252916 | **-10.53631** | 1.11E-07 | -10.5364 | 3.35E-06 | -7.291204 | 2.9614583 | -10.47212 | 0.1028446 |
| F1 | 7.57E-101 | 1.30E-100 | 2.71E-176 | 0 | 4.60E-95 | 1.03E-94 | 3.80E-97 | 4.66E-97 | **0** | **0** |
| F2 | 8.48E-53 | 1.75E-52 | 3.12E-90 | 6.92E-90 | 8.43E-53 | 1.88E-52 | 1.02E-52 | 2.14E-52 | **1.63E-177** | **0** |
| F3 | 9.74E-81 | 1.37E-80 | 6.34E-142 | 1.42E-141 | 2.74E-86 | 6.12E-86 | 3.89E-86 | 8.67E-86 | **2.73E-198** | **0** |
| F4 | 1.04E-49 | 2.23E-49 | 1.71E-89 | 3.69E-89 | 2.64E-50 | 5.72E-50 | 9.48E-51 | 1.55E-50 | **1.50E-188** | **0** |
| F5 | 0.01221134 | 0.015032 | 1.624055 | 1.523146 | 0.012310 | 0.013932 | 0.018180 | 0.022361 | **0.0034244** | **0.01505** |
| F6 | 0.0001033 | 0.000105 | 0.018239 | 0.023785 | 7.76E-05 | 7.82E-05 | 0.000213 | 0.000138 | 0.000198 | 0.00031 |
| F7 | 0.00011856 | 7.99E-05 | 0.000142 | 0.000184 | 0.000154 | 6.13E-05 | 5.52E-05 | 6.33E-05 | **8.36E-07** | **4.60E-07** |
| F8 | -12568.823 | 1.0914916 | -11258.41 | 1137.674 | -12568.6 | 0.442414 | -12568.62 | 0.699447 | -12930.83 | 0.88076 |
| F9 | **0** | **0** | **0** | **0** | **0** | **0** | **0** | **0** | **0** | **0** |
| F10 | **8.88E-16** | **0** | **8.88E-16** | **0** | **8.88E-16** | **0** | **8.88E-16** | **0** | **8.88E-16** | **0** |
| F11 | **0** | **0** | **0** | **0** | **0** | **0** | **0** | **0** | **0** | **0** |
| F12 | 6.97E-06 | 5.69E-06 | 0.0004790 | 0.000367 | 7.69E-06 | 6.19E-06 | 5.71E-06 | 6.59E-06 | **8.46E-06** | **1.29E-05** |
| F13 | 6.66E-05 | 7.86E-05 | 0.0108518 | 0.010619 | 2.31E-05 | 1.89E-05 | 5.42E-05 | 5.10E-05 | 3.30E-05 | 2.44E-05 |
| F14 | 1.1968092 | 0.4445424 | 0.9980058 | 4.56E-06 | 0.998003 | 7.47E-11 | 2.1829775 | 2.137783 | **0.99901** | **0.444542** |
| f15 | 0.0003103 | 3.28E-06 | 0.0004604 | 0.000170 | 0.000311 | 2.99E-06 | 0.0003194 | 2.76E-06 | 0.0003059 | 1.00E-05 |
| F16 | **-1.031628** | 3.19E-10 | **-1.031628** | 2.87E-10 | **-1.031628** | 6.99E-10 | **-1.031628** | 1.04E-08 | **-1.031628** | 1.60E-10 |
| F17 | 0.39788795 | 1.19E-06 | 0.397978 | 0.000142 | 0.397887 | 1.33E-07 | 0.3978874 | 1.14E-07 | 0.3978881 | 1.55E-06 |
| F18 | 13.800001 | 14.788511 | 3.000018 | 2.41E-05 | 8.400003 | 12.07477 | 8.400002 | 12.07476 | 3.000002 | 2.93E-06 |
| F19 | -3.8527389 | 4.77E-05 | -3.762825 | 0.112669 | **-3.86278** | 3.36E-07 | -3.862606 | 0.000367 | -3.859995 | 0.006209 |
| F20 | -3.2701541 | 0.071042 | -3.31470 | 0.005088 | -3.27404 | 0.065469 | -3.296126 | 0.057369 | -3.248782 | 0.066768 |
| F21 | -6.0744263 | 2.279263 | -6.056752 | 2.266704 | -5.05517 | 1.96E-05 | -5.055163 | 2.79E-05 | -10.15312 | 8.86E-05 |
| F22 | -5.0876202 | 3.99E-05 | -5.079582 | 0.009285 | -5.08761 | 4.06E-05 | -5.087622 | 3.21E-05 | -10.40272 | 0.000171 |
| F23 | -5.1284078 | 9.20E-05 | -6.201579 | 2.422959 | -5.12844 | 2.67E-05 | -3.896684 | 1.707097 | -10.53621 | 0.000113 |

**4.1.2. Evaluation of exploration capability (F8-F23).** The multimodal functions F8-F23 contain a large number of local optimal values, which increase exponentially with increasing dimension. Therefore, these functions are suitable for evaluating the exploration capability and the ability to avoid local optima. It can be seen from Table 1 and Figs 5–7 that the HHO-CS-OELM algorithm outperforms the other algorithms in most of the multimodal functions F8-F13. For F8, the HHO-CS-OELM algorithm is inferior only to the ABC algorithm, but superior to all other algorithms. For F9-F11, although most algorithms can obtain the optimal solutions, the convergence speed of the HHO-CS-OELM algorithm is the fastest. For F12, the HHO-CS-OELM algorithm is inferior only to SFLA algorithm, but superior to all other algorithms. For F13, the HHO-CS-OELM algorithm is inferior only to the SFLA, DEHHO and CHHO algorithms, but superior to all other algorithms. However, when compared to the standard HHO algorithm alone, the HHO-CS-OELM algorithm is a winner in all conditions, which indicates that HHO-CS-OELM is completely superior to the HHO algorithm.

For F14, the HHO-CS-OELM algorithm is superior to all other algorithms, and the convergence speed is also the fastest. For F15, the HHO-CS-OELM algorithm is inferior to only the BHHO algorithm. For F16, the HHO-CS-OELM algorithm is inferior to only the SFLA, ABC and DEHHO algorithms, but superior to all other algorithms. For F17, however, the HHO-CS-OELM algorithm is superior to only the HHO and HHHO algorithms, but inferior to all

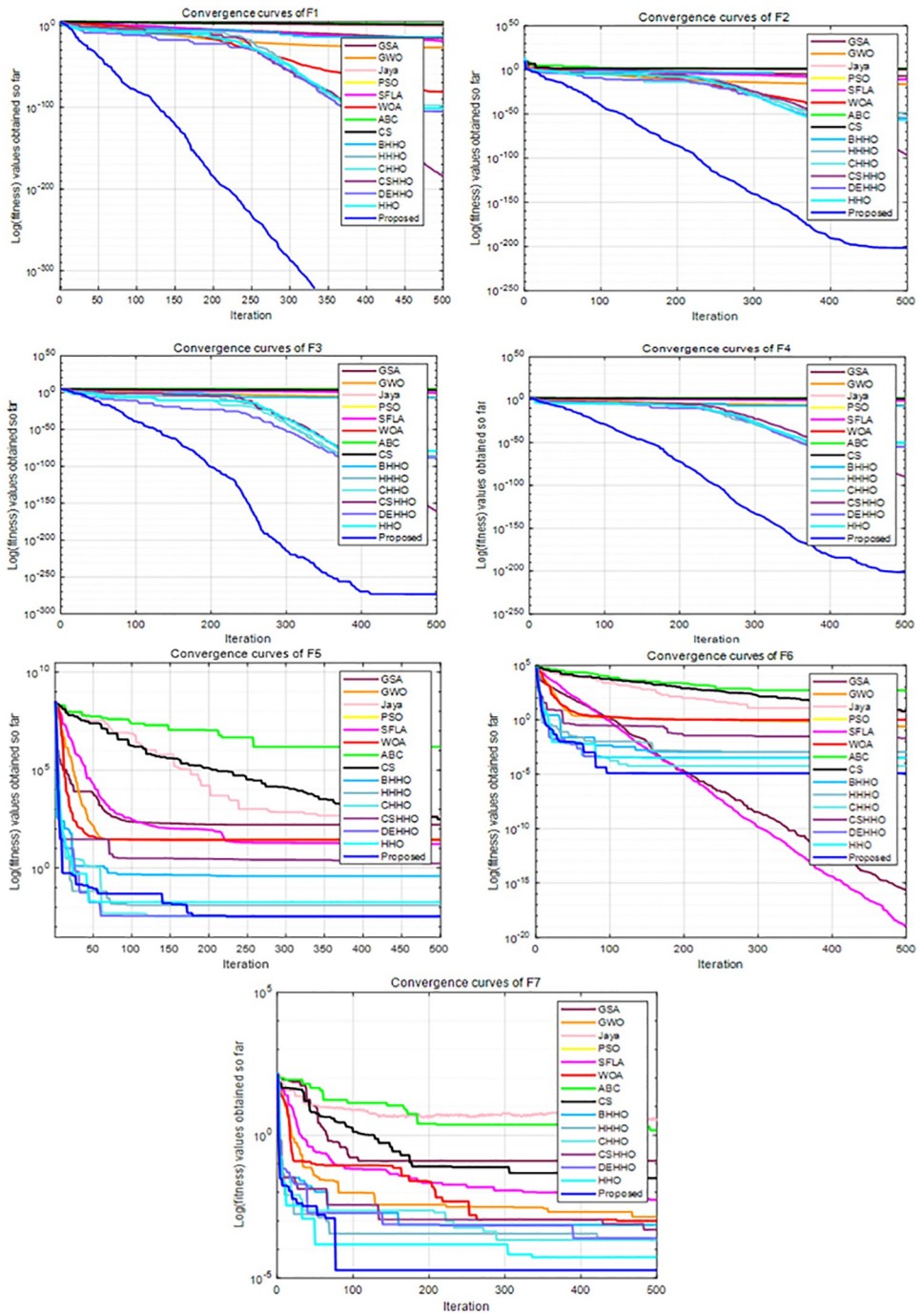

**Fig 5. Convergence curves of F1-F7.**

other algorithms. For F18, the HHO-CS-OELM algorithm is inferior to only the SFLA algorithm, but superior to all other algorithms. For F19, the HHO-CS-OELM algorithm is superior to only the HHO and BHHO algorithms, but inferior to all other algorithms. For F20, the HHO-CS-OELM algorithm is inferior to only the WOA and CS algorithms, but superior to all other algorithms. For F21-F23, the HHO-CS-OELM algorithm is inferior to only the ABC

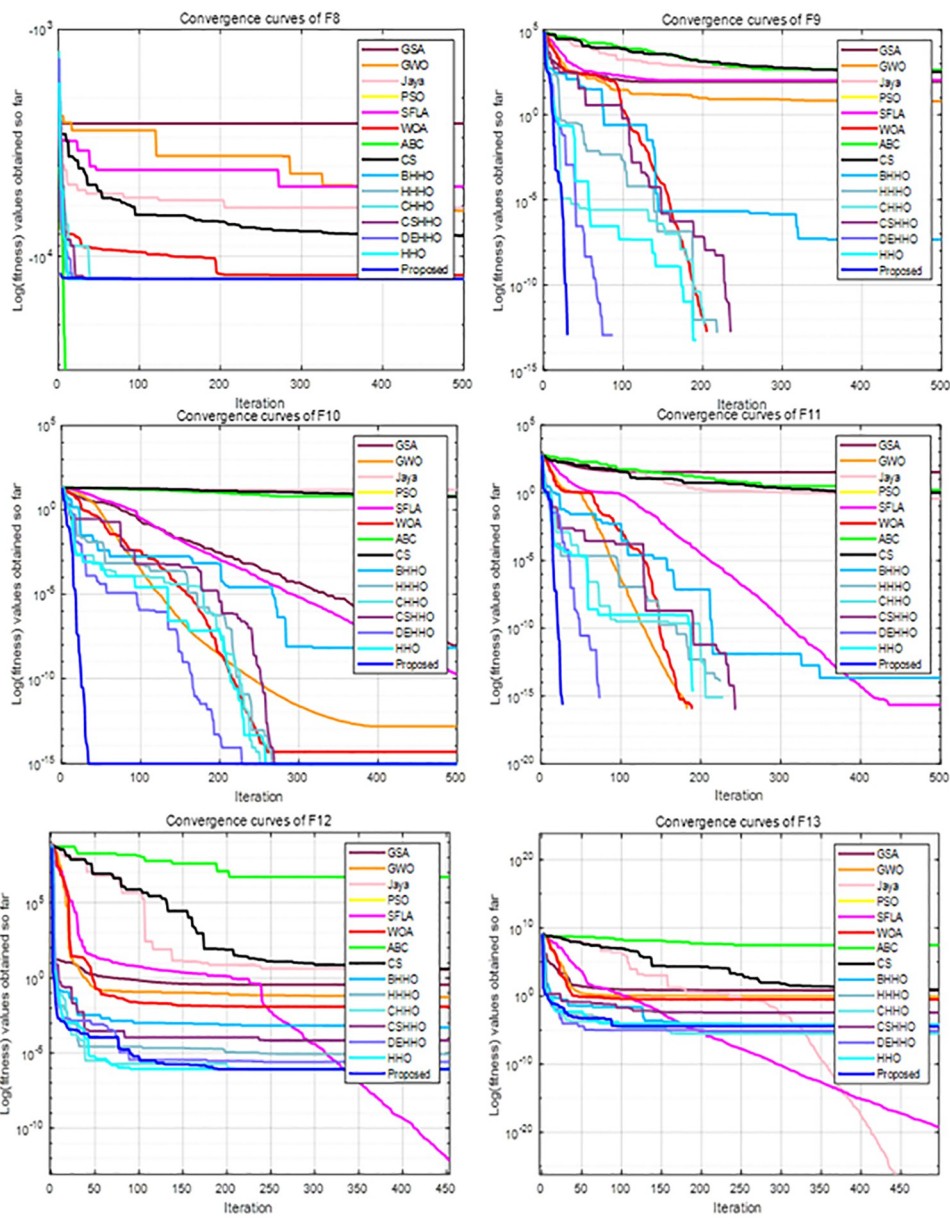

**Fig 6. Convergence curves of F8-F13.**

algorithm and superior to all other algorithms. For F14-F23, the HHO-CS-OELM algorithm performs barely satisfactory with respect to all other algorithms; however, it still completely outperforms the standard HHO algorithm. In summary, these results show that HHO-CS-OELM can provide superior exploration capability.

In addition, the proposed algorithm can obtain optimal values for 11 functions out of 23 functions; while it outperforms the standard HHO algorithm on all of the 23 functions. Therefore, the above results reveal that the chaotic sequence and opposite elite learning mechanism can effectively balance the exploitation and exploration capabilities and improve the performance of the HHO algorithm.

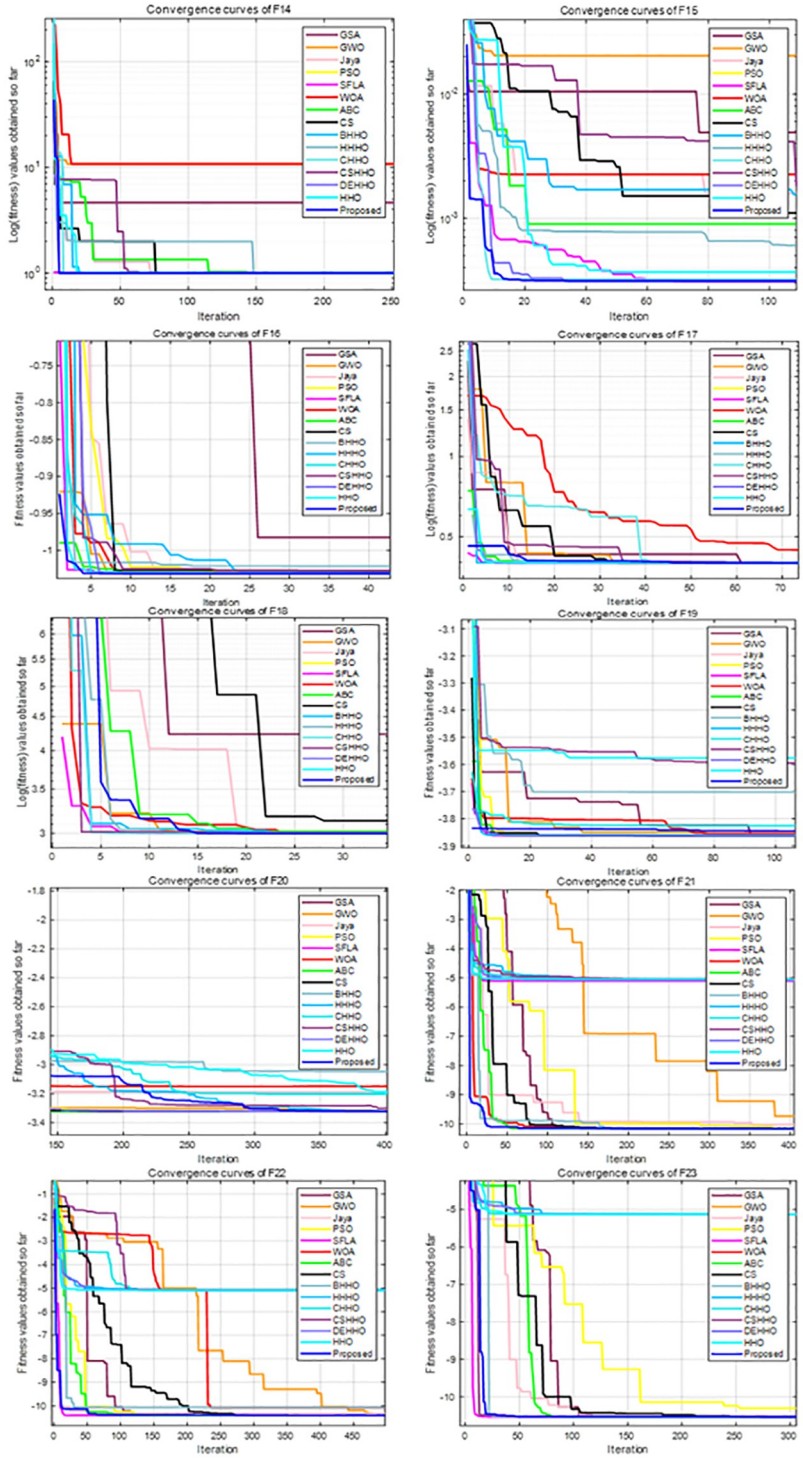

**Fig 7. Convergence curves of F14–F23.**

## 4.2 Engineering application

In this section, an engineering problem on indoor positioning based on the improved BP neural network, is performed to verify the performance of the proposed method. A BP neural

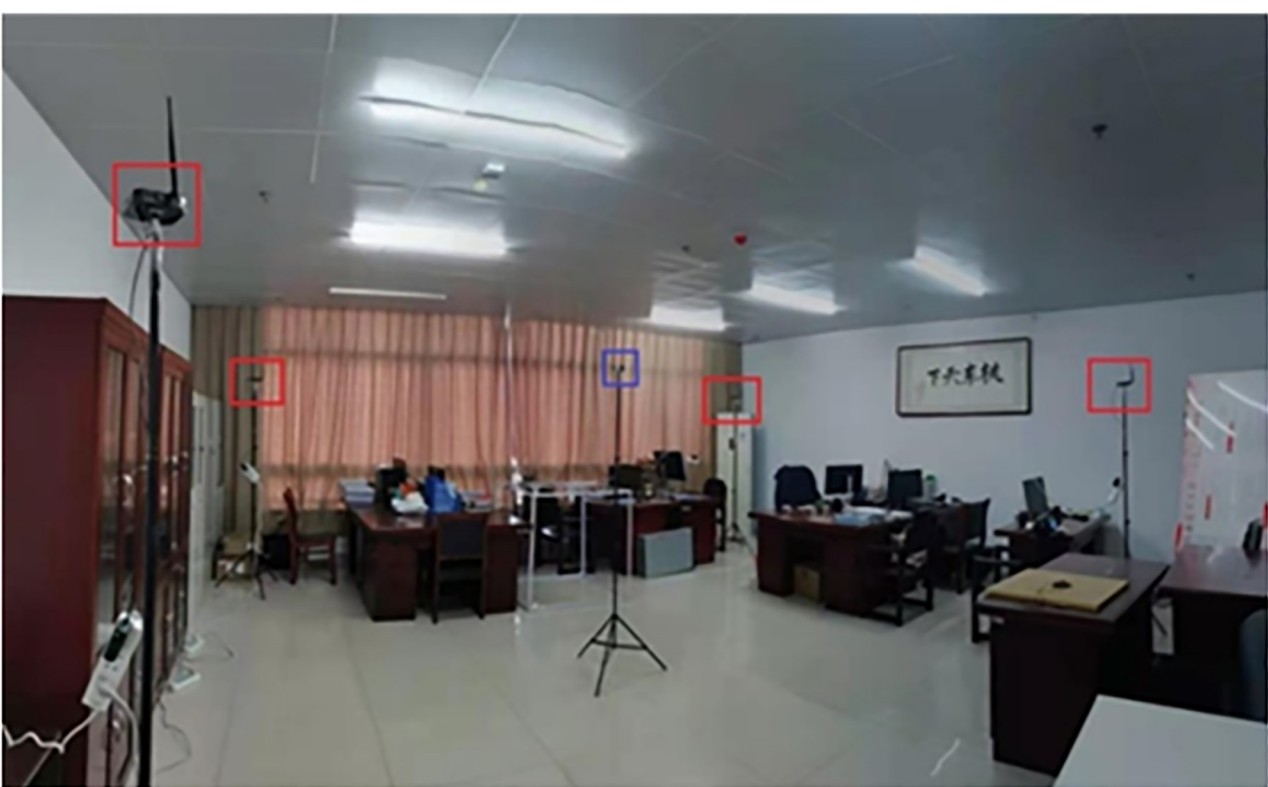

**Fig 8. The positioning scenario of UWB.**

network is a multilayer feedforward neural network equipped with error back propagation. The BP neural network is one of the most widely used artificial neural network models at present. It is composed of input layer, hidden layer and output layers, for which, the number of neurons in each layer can be set randomly according to requirements, while the performance of the network varies with the number of different structures. However, the initial thresholds and weights of the network generated randomly can easily steer the network into a local extrema. Therefore, to alleviate this shortcoming, many intelligent optimization algorithms are used to optimize the initial thresholds and weights [45–48].

Thus, in this section, HHO, BHHO, HHHO, CHHO, CSHHO, DEHHO and the proposed improved HHO algorithms are applied to optimize the initial thresholds and weights, and then the improved BP neural network is applied to improve the accuracy of the ultrawide band (UWB) positioning system.

The indoor positioning system-based UWB is configured with four UWB base stations deployed at four corners of a 5.6 m x 4.8 m room, which is shown in (Fig 8). The red rectangle is the base station, and the blue rectangle is the location tag. In the offline training stage of the BP neural network, 64 sample points are sampled, with (1.0 m, 1.0 m) as the starting point, at intervals of 40 cm, to be used as the training dataset. As shown in (Fig 9), in the online test stage of BP neural network, 49 sample points are sampled with (1.2 m, 1.2 m) as the starting point, at intervals of 40 cm, to be used as the test dataset; where the red circle indicates the measured position and the blue asterisk denotes the real position.

According to the experimental configuration, the structure of the BP network consists of 2 input and 2 output neurons, and 7 neurons in hidden layer. The activation function is set as a

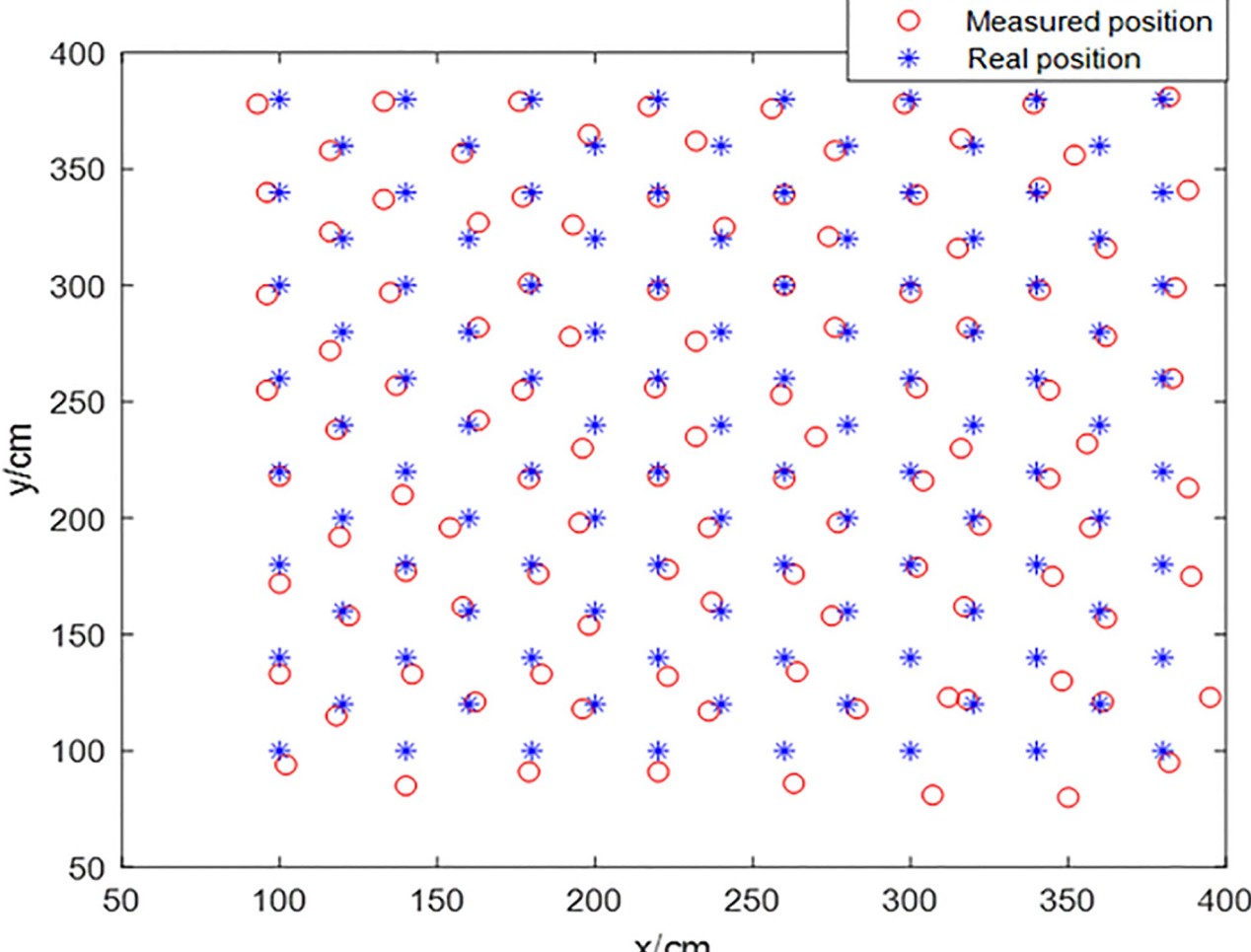

**Fig 9. Measured positions and real positions.**

Sigmoid function in hidden layer, transfer function is set as a Purelin function in the output layer, gradient descending method is used as the training method, mean square error function is used as the performance index, and the maximum training time is 2000, the learning rate is 0.01, and the initial weights and thresholds are generated by a random generation method and the swarm intelligent optimization algorithms, respectively. (Figs 10 to 13) show the experimental results.

(Fig 10) shows that the proposed HHO-CS-OELM algorithm has the fastest convergence speed. In (Fig 11), the black pillar indicates the average positioning error of the BP neural network, when random weights and thresholds were 7.76 cm, which demonstrates the worst performance. On the other hand, the red pillar indicates that the resulting average error of the improved BP neural network optimized by the HHO-CS-OELM algorithm is 3.81 cm, which is the best performance. These results reveal that the improved BP neural network can improve the performance of the BP neural network.

In (Fig 12), the red line represents the positioning error of the improved BP neural network optimized by the HHO-CS-OELM algorithm, while the black line denotes the measuring error. In Fig 13, the green point, the red circle, and the blue asterisk indicate the predicted

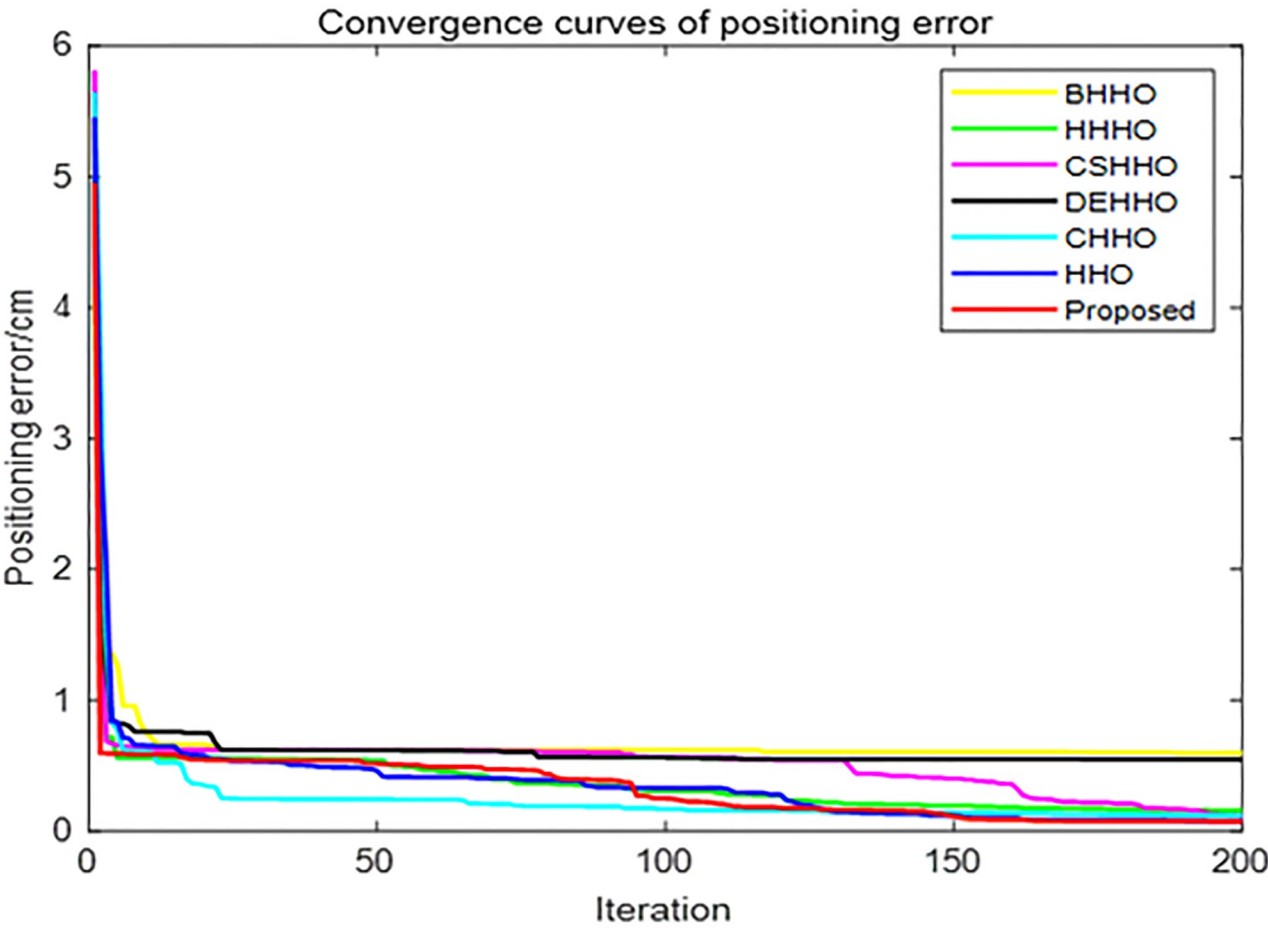

**Fig 10. Convergence curves of different algorithms.**

positions obtained by the BP neural network and the HHO-CS-OELM algorithm along with the measured positions and real positions, respectively. Figs 12 and 13 show that the positioning errors of approximately 67.35% (33/49) of the test points are significantly improved by the modified BP neural network. These results reveal that the HHO-CS-OELM can significantly improve the performance of the BP neural network and further verify that the two strategies can effectively enhance the performance of the HHO algorithm in this paper.

## 5 Conclusions and future works

In this paper, we propose a new HHO variant algorithm based on chaotic mapping and an opposite elite learning mechanisms. The chaotic sequence recombination mechanism is applied at the initial population stage to improve the diversity of the population and enhance the exploration ability of the HHO algorithm. The opposite elite learning recombination mechanism is used at the last stage of each iteration, to effectively maintain the optimal individual and enhance the exploitation ability. On the other hand, this method can overcome the shortcoming at the late iteration in the HHO algorithm: inability to perform global search. In addition, the exploitation and exploration capabilities of the HHO algorithm are balanced. The performance of the HHO-CS-OELM algorithm is verified by comparison with 14 optimization algorithms on 23 benchmark functions and an engineering problem which aims to optimize

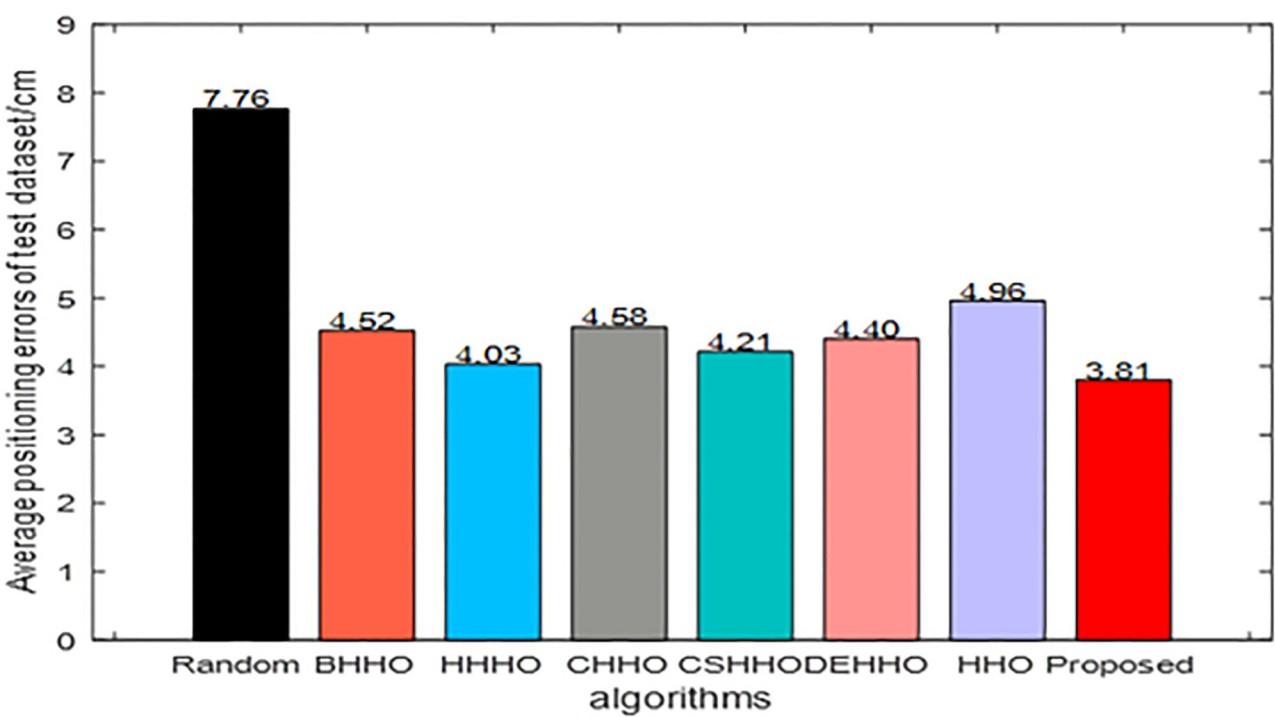

**Fig 11. Average positioning errors of the test dataset.**

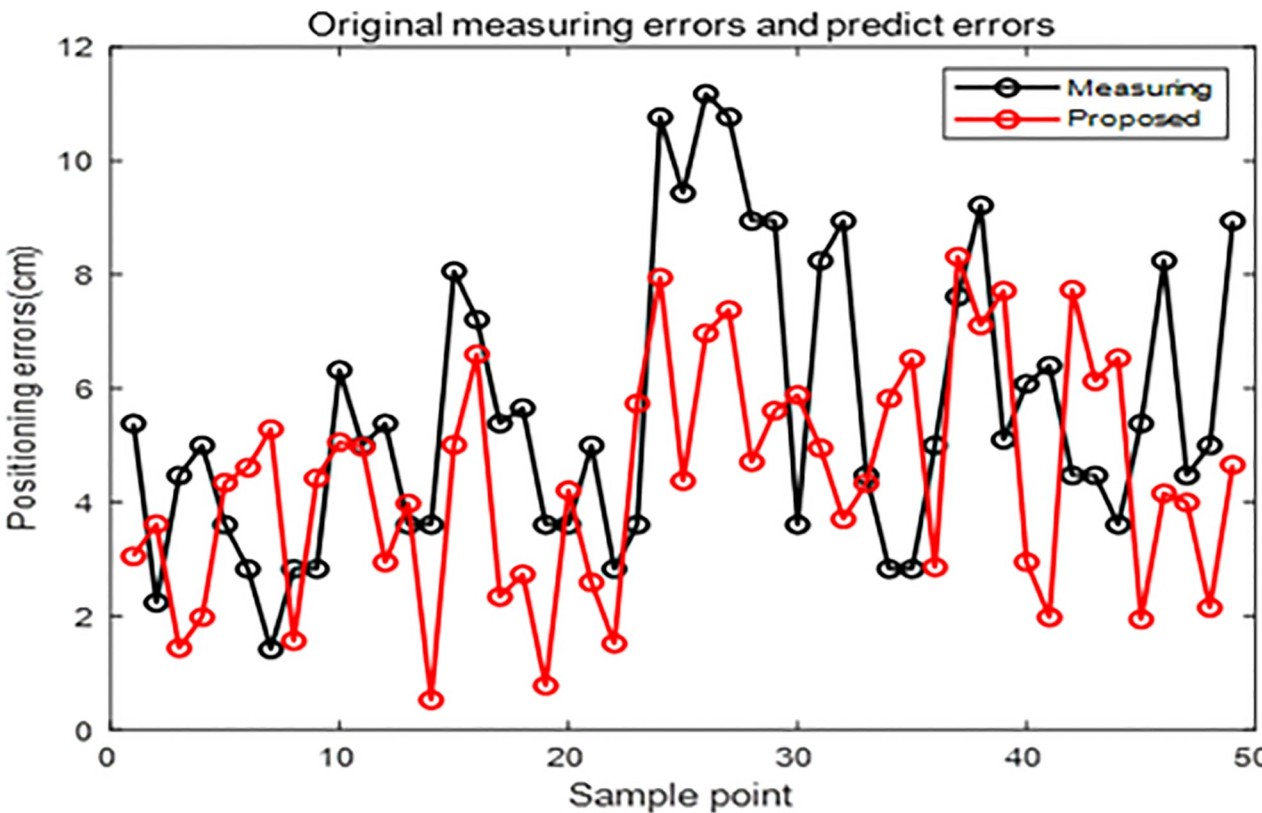

**Fig 12. Measuring and predicting errors of the test dataset.**

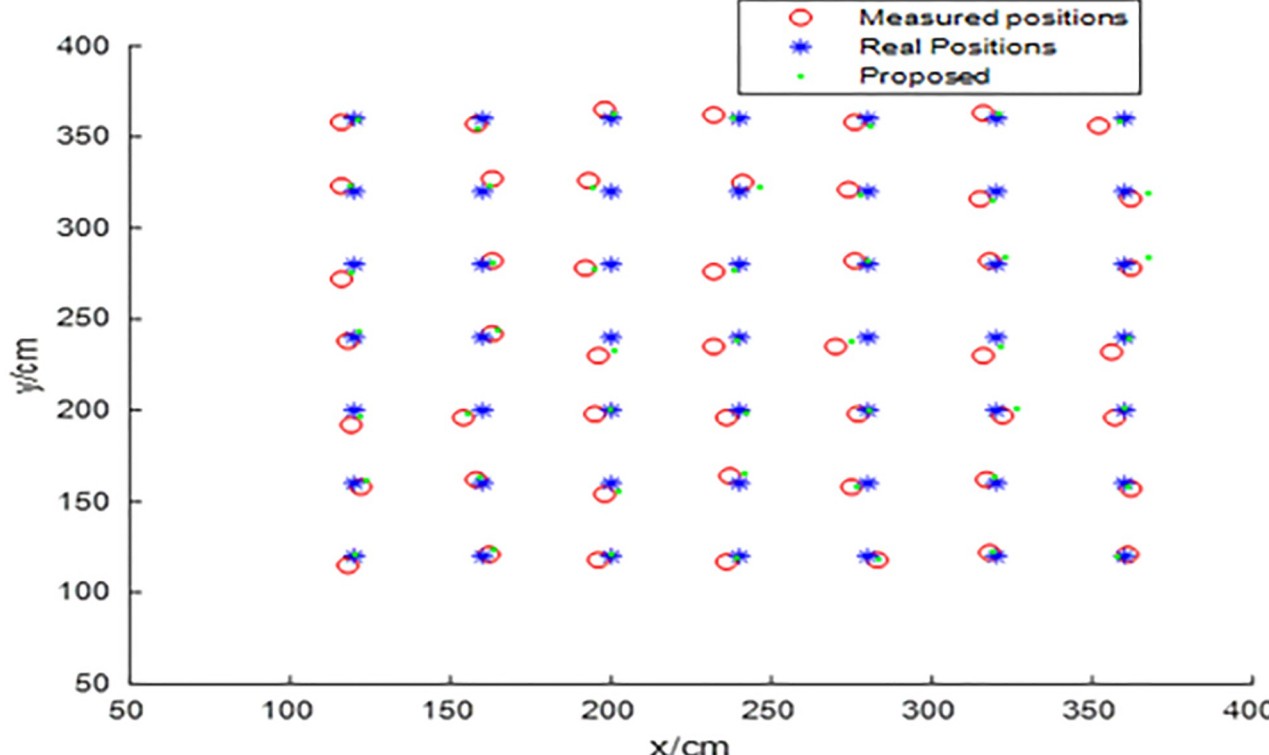

**Fig 13. Predicted, real and measured positions.**

the weights and thresholds of the BP neural network for indoor positioning. Experimental analyses revealed that the proposed algorithm can obtain optimal values for 11 functions out of 23 functions compared to other state-of-the-art SI algorithms, while it outperforms the standard HHO algorithm on all of the 23 functions, so the HHO-CS-OELM algorithm offers competitive results compared to other state-of-the-art SI algorithms, and verified that the HHO-CS-OELM algorithm has better performance than others. The improved BP neural network optimized by the HHO-CS-OELM algorithm reduced the indoor average positioning error from 7.76 cm to 3.81 cm which is obtained by BP neural network, it also revealed that the indoor positioning accuracy obtained by using the BP neural network with the proposed algorithm is significantly improved, which further verifies the superiority of the HHO-CS-OELM algorithm.

To further improve the performance of the HHO-CS-OELM algorithm, outcomes of other benchmark functions still need to be enhanced, and the exploitation and exploration capabilities of the HHO-CS-OELM algorithm need to be investigated further. Moreover, the HHO-CS-OELM algorithm should be used to solve the system optimal parameters, optimal parameters of neural network model, and so on.

## Acknowledgments

We truly appreciate the Intelligent Network Automobile Laboratory of West Anhui University in Lu'an for providing experimental equipment. We are also grateful to Professor Jie Fang for his guidance.

## Author Contributions

**Investigation:** Zhengyu Liu.

**Methodology:** Jie Fang.

**Validation:** Yu Liu.

**Writing – original draft:** Ting Yang, Chaochuan Jia.

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
