## [Decision Letter · Decision Letter 0]

29 Apr 2022

PONE-D-22-04874An Improved Harris Hawks Optimization Algorithm based on Chaotic Sequence and Opposite Elite Learning MechanismPLOS ONE

Dear Dr. Jia,

Thank you for submitting your manuscript to PLOS ONE. After careful consideration, we feel that it has merit but does not fully meet PLOS ONE’s publication criteria as it currently stands. Therefore, we invite you to submit a revised version of the manuscript that addresses the points raised during the review process.

We look forward to receiving your revised manuscript.

Kind regards,

Pijush Samui

Academic Editor

PLOS ONE

Journal Requirements:

"This work was partially supported by the Natural Science Research Project of the universities in Anhui Province (NO. KJ2021A0953), and Natural Science Key Scientific Research Project of West Anhui University (NO. 0041021003, WXZR201903)."

We note that you have provided funding information. However, funding information should not appear in the Funding section or other areas of your manuscript. We will only publish funding information present in the Funding Statement section of the online submission form. 

Please remove any funding-related text from the manuscript and let us know how you would like "This work was partially supported by Natural Science research project of Universities in Anhui Province(NO.KJ2021A0953), Natural Science Key Scientific Research Project of West Anhui University (NO. 0041021003, WXZR201903)."

Reviewers' comments:

Reviewer's Responses to Questions

**Comments to the Author**

1. Is the manuscript technically sound, and do the data support the conclusions?

Reviewer #1: Yes

Reviewer #2: No

2. Has the statistical analysis been performed appropriately and rigorously? 

Reviewer #1: Yes

Reviewer #2: No

3. Have the authors made all data underlying the findings in their manuscript fully available?

Reviewer #1: Yes

Reviewer #2: No

4. Is the manuscript presented in an intelligible fashion and written in standard English?

Reviewer #1: Yes

Reviewer #2: No

5. Review Comments to the Author

Reviewer #1: 1. Please complete the Abstract within 200 words and it should be very precise.

2. Motivations of the paper are not clear.

3. Disadvantages of the existing schemes must be discussed. It is important to justify and motivate the new work.

4. In the Introduction, there must be point-wise contributions.

5. The authors have not given Related Works section, so the Introduction section must be very strong.

6. Section 2 must be renamed as “Background Studies”.

7. What are the major advantages of Harris hawk optimization? Mention it.

8. In Eq. (3), how the value of T is changed?

9. Rename section 3 as “Proposed Scheme”. Moreover, motivations must be mentioned.

10. What are the used of Opposite Elite Learning Recombination Mechanism?

11. It is difficult to identify the novelty of the proposed scheme.

12. The authors must include a section as “Experimental Environment”.

13. Discussion on figure 4 must be given.

14. How the benchmark functions are chosen?

15. Table 1 is too long. Moreover, it is not clear how the results of Table 1 are generated.

16. Section 5 must be renamed as Conclusions and Future Works

17. Applications of the proposed scheme can be discussed in the last section.

18. All the figures, tables, equations and references must be cited in the text.

19. Key terms of the equations must be defined.

20. Improve the English language

21. Include the following references to improve the reference section:

• Robust optimisation algorithm for the measurement matrix in compressed sensing. CAAI Transactions on Intelligence Technology. DOI: https://doi.org/10.1049/trit.2018.1011

• Ensemble algorithm using transfer learning for sheep breed classification. DOI: 10.1109/SACI51354.2021.9465609

• Image-denoising algorithm based on improved K-singular value decomposition and atom optimization. CAAI Transactions on Intelligence Technology. DOI: https://doi.org/10.1049/cit2.12044

• IFODPSO-based multi-level image segmentation scheme aided with Masi entropy. Journal of Ambient Intelligence and Humanized Computing. DOI: https://doi.org/10.1007/s12652-020-02506-w

• Adaptive multifactorial particle swarm optimisation. CAAI Transactions on Intelligence Technology. DOI: https://doi.org/10.1049/trit.2018.1090

• A boosting-aided adaptive cluster-based undersampling approach for treatment of class imbalance problem. International Journal of Data Warehousing and Mining (IJDWM). DOI: 10.4018/IJDWM.2020070104

Reviewer #2: It is interesting article to fix the local convergence. The following reasons to me for rejection:

1. English is the main problem, flow specially in related work is missing.

2. Methodology is not much clear, how the minimum convergence obtained or how global minimum will help to address this concern.

3. Fitness function is not clear How E is updated.

4. Results are not compared with the existing models. Even datasets is also limited.

5. Novelty is limited

6. PLOS authors have the option to publish the peer review history of their article (what does this mean?). If published, this will include your full peer review and any attached files.

Reviewer #1: No

Reviewer #2: **Yes: **Dr. Krishan Kumar

---

## [Author Response · Author response to Decision Letter 0]

9 Jun 2022

These are my responses to each question. Thank you for your advice in your busy schedule, I thought about each question carefully and gave an explanation. Due to my limited research skills, there may be some incorrect answers, please criticize and correct me. Thanks again! Best wishes to you!

Reviewer #1:

1. Please complete the Abstract within 200 words and it should be very precise.

Reply: It has been modified, now there is only 181 words.

2. Motivations of the paper are not clear.

Reply：The HHO still has some limitations, such as the multiplicity of solutions generated by a randomized policy that is finite in the initialization phase. Moreover, because global exploration is only performed in the first half of the iteration, it is difficult to balance the global exploration and local exploitation capacities by using the escaping energy of prey, so the algorithm may converge slowly, has low solution accuracy and prematurely falls into a local optimal solution. To conquer the limitations of HHO, an improved HHO algorithm has been proposed in this paper.

3. Disadvantages of the existing schemes must be discussed. It is important to justify and motivate the new work.

Reply：HHO is a new swarm intelligence optimization algorithm proposed by Heidari et al.[9] in 2019 that mimics the way Harris eagles find and chase prey in nature, including global exploration, local besiege and pounce behaviour. However, similar to the other SI algorithms, the HHO still has some limitations, such as the multiplicity of solutions generated by a randomized policy that is finite in the initialization phase. Moreover, because global exploration is only performed in the first half of the iteration, it is difficult to balance the global exploration and local exploitation capacities by using the escaping energy of prey, so the algorithm may converge slowly, have low solution accuracy and prematurely fall into a local optimal solution.

4. In the Introduction, there must be point-wise contributions.

Reply：the contributions of this paper are as follows:(i) a chaotic sequence chaotic sequence recombination mechanism (CSRM) strategy is proposed, which enhances the distribution of the initialized solutions in the search space, and accelerates the convergence rate of HHO;(ii) the generalized opposition-based learning recombination mechanism (OBLRM) is proposed, which can have the opportunity to carry out global search in the later period of iteration to jump out of the local optimum and improve the accuracy of the solution.

5. The authors have not given Related Works section, so the Introduction section must be very strong.

Reply：In the introduction section, firstly, the development and application of optimization algorithm are described, secondly, the research situation of HHO algorithm and many HHO variants are summarized, finally, the shortcomings of HHO algorithm are summarized and the improved methods and contributions of this paper are proposed. Related work is integrated into the introduction section, and 47 references have been cited in this paper.

6. Section 2 must be renamed as “Background Studies”.

Reply：it has been modified

7. What are the major advantages of Harris hawk optimization? Mention it.

Reply: HHO is a new swarm intelligence optimization algorithm proposed by Heidari et al.[9] in 2019 that mimics the way Harris eagles find and chase prey in nature, which has strong global search capability and adjusts few parameters. It has been modified.

8. In Eq. (3), how the value of T is changed?

Reply: T denotes the maximum number of iterations, which is defined by experience. 

9. Rename section 3 as “Proposed Scheme”. Moreover, motivations must be mentioned.

Reply: it has been modified in section 3. To improve the diversity of initial population and enhance the ability to jump out of local optimal solution of HHO algorithm, the specific implementation of the proposed algorithm is described in detail in this section. Two recombination mechanisms, the chaotic sequence recombination mechanism (CSRM) and generalized opposition-based learning recombination mechanism (OBLRM), are introduced to enhance the performance of the HHO algorithm. The improved HHO algorithm does not change the structure of the HHO. Fig. 3 shows the optimization process of the proposed algorithm.

10. What are the used of Opposite Elite Learning Recombination Mechanism?

Reply: It is well known that in the HHO algorithm, even if the selected region is not globally optimal, the global search is no longer carried out in the later periods of iteration, so the HHO tends to converge to the local optimum prematurely. However, when the opposite elite learning recombination mechanism is embedded in the HHO algorithm, the improved algorithm can have the opportunity to carry out global search in the later period of iteration to jump out of the local optimum and improve the accuracy of the solution. It is shown in section 3.2.

11. It is difficult to identify the novelty of the proposed scheme.

Reply: HHO is a new swarm intelligence optimization algorithm proposed by Heidari et al.[9] in 2019 that mimics the way Harris eagles find and chase prey in nature, including global exploration, local besiege and pounce behaviour. However, similar to the other SI algorithms, the HHO still has some limitations, such as the multiplicity of solutions generated by a randomized policy that is finite in the initialization phase. Moreover, because global exploration is only performed in the first half of the iteration, it is difficult to balance the global exploration and local exploitation capacities by using the escaping energy of prey, so the algorithm may converge slowly, has low solution accuracy and prematurely falls into a local optimal solution. In this paper, to conquer the limitations of HHO, a chaotic sequence chaotic sequence recombination mechanism (CSRM) strategy is proposed, which enhances the distribution of the initialized solutions in the search space, and accelerates the convergence rate of HHO, and the generalized opposition-based learning recombination mechanism (OBLRM) is proposed, which can have the opportunity to carry out global search in the later period of iteration to jump out of the local optimum and improve the accuracy of the solution. 

12. The authors must include a section as “Experimental Environment”.

Reply：In section 4, the experimental environment is given. All experiments are carried out on a Windows 10 operating system with MATLAB R2019a on a PC with Inter(R) core i7-10750H and 16 GB RAM memory.

13. Discussion on figure 4 must be given.

Reply：Fig.4(a) and (b) are unimodal functions which have only one minimum value, however, from Fig.4(c) to Fig.4(f) are multimodal functions which have a lot of local minimum values.

14. How the benchmark functions are chosen?

Reply：23 benchmark functions are shown in reference 6, which are categorised as unimodal, multimodal and fixed dimension multimodal, and they are specifically designed to test optimization algorithms. It has been referenced in the article.

15. Table 1 is too long. Moreover, it is not clear how the results of Table 1 are generated.

Reply：In order to fully verify the performance of the improved algorithm in this paper, the results of 23 benchmark functions optimized by 15 optimization algorithms are given in Table 1. Due to the limitation of page scope, it can only be decomposed into several sub-tables. However, we analyze the results in the table from two perspectives. (1) Evaluation of exploitation capability (F1-F7), For all unimodal functions excluding F6, the proposed algorithm acquires the best optimal average values and standard deviations, which indicates that the accuracy and stability of the proposed algorithm are the best among 15 optimization algorithms. (2) Evaluation of exploration capability (F8-F23), It can be seen from Table 1 that the proposed algorithm outperforms the other algorithms in most of the multimodal functions F8-F13. For F8, the proposed algorithm is inferior only to the ABC algorithm, but superior to all other algorithms. For F9-F11, although most algorithms can obtain the optimal solutions, the convergence speed of the proposed algorithm is the fastest. For F12, the proposed algorithm is inferior only to SFLA algorithm, but superior to all other algorithms. For F13, the proposed algorithm is inferior only to the SFLA, DEHHO and CHHO algorithms, but superior to all other algorithms. However, when compared to the standard HHO algorithm alone, the proposed algorithm is a winner in all conditions, which indicates that proposed is completely superior to the HHO algorithm. For F14, the proposed algorithm is superior to all other algorithms, and the convergence speed is also the fastest. For F15, the PROPOSED algorithm is inferior to only the BHHO algorithm. For F16, the proposed algorithm is inferior to only the SFLA, ABC and DEHHO algorithms, but superior to all other algorithms. For F17, however, the proposed algorithm is superior to only the HHO and HHHO algorithms, but inferior to all other algorithms. For F18, the proposed algorithm is inferior to only the SFLA algorithm, but superior to all other algorithms. For F19, the proposed algorithm is superior to only the HHO and BHHO algorithms, but inferior to all other algorithms. For F20, the proposed algorithm is inferior to only the WOA and CS algorithms, but superior to all other algorithms. For F21-F23, the proposed algorithm is inferior to only the ABC algorithm and superior to all other algorithms. For F14-F23, the proposed algorithm performs barely satisfactory with respect to all other algorithms; however, it still completely outperforms the standard HHO algorithm. In summary, these results show that proposed can provide superior exploration capability. In addition. the proposed algorithm can obtain optimal values for 11 functions out of 23 functions; while it outperforms the standard HHO algorithm on all of the 23 functions. Therefore, the above results reveal that the chaotic sequence and opposite elite learning mechanism can effectively balance the exploitation and exploration capabilities and improve the performance of the HHO algorithm. 

16. Section 5 must be renamed as Conclusions and Future Works

Reply：it has been modified.

17. Applications of the proposed scheme can be discussed in the last section.

Reply：it has been modified.

18. All the figures, tables, equations and references must be cited in the text.

Reply：it has been modified. Fig. 1. Different phases of HHO (Heidari et al.[9]). Fig. 2. Optimization process of basic HHO (Heidari et al.[9]). 

19. Key terms of the equations must be defined.

Reply：it has been modified. 

20. Improve the English language

Reply: it has been modified by AJE, and the verification code is E65D-842F-B583-0487-1D9C.

21. Include the following references to improve the reference section:

Reply：it has been modified. The following references have been added to the reference section.

• Robust optimization algorithm for the measurement matrix in compressed sensing. CAAI Transactions on Intelligence Technology. DOI: https://doi.org/10.1049/trit.2018.1011

• Ensemble algorithm using transfer learning for sheep breed classification. DOI: 10.1109/SACI51354.2021.9465609

• Image-denoising algorithm based on improved K-singular value decomposition and atom optimization. CAAI Transactions on Intelligence Technology. DOI: https://doi.org/10.1049/cit2.12044

• IFODPSO-based multi-level image segmentation scheme aided with Masi entropy. Journal of Ambient Intelligence and Humanized Computing. DOI: https://doi.org/10.1007/s12652-020-02506-w

• Adaptive multifactorial particle swarm optimization. CAAI Transactions on Intelligence Technology. DOI: https://doi.org/10.1049/trit.2018.1090

• A boosting-aided adaptive cluster-based undersampling approach for treatment of class imbalance problem. International Journal of Data Warehousing and Mining (IJDWM). DOI: 10.4018/IJDWM.2020070104

Reviewer #2: 

1. English is the main problem, flow specially in related work is missing.

Reply: It has been modified by AJE, and the verification code is E65D-842F-B583-0487-1D9C.

In the introduction section, firstly, the development and application of optimization algorithm are described, secondly, the research situation of HHO algorithm and many HHO variants are summarized, finally, the shortcomings of HHO algorithm are summarized and the improved methods and contributions of this paper are proposed. Related work is integrated into the introduction section, and 47 references have been cited in this paper.

2. Methodology is not much clear, how the minimum convergence obtained or how global minimum will help to address this concern.

Reply：In this paper, to conquer the limitations of HHO, a chaotic sequence chaotic sequence recombination mechanism (CSRM) strategy is proposed, which enhances the distribution of the initialized solutions in the search space, and accelerates the convergence rate of HHO, and the generalized opposition-based learning recombination mechanism (OBLRM) is proposed, which can have the opportunity to carry out global search in the later period of iteration to jump out of the local optimum and improve the accuracy of the solution. 

3. Fitness function is not clear How E is updated.

Reply：E is updated according to equation 3 in this paper.

4. Results are not compared with the existing models. Even datasets is also limited.

Reply：The improved algorithm has been compared with 14 optimization algorithms on the 23 benchmark functions, the results show that the improved algorithm has obvious advantages, indoor positioning experiment can further prove that the performance of original HHO algorithm has been greatly improved, the dataset is measured by ourselves rather than the public dataset. 

5. Novelty is limited

Reply：This is an important comment, HHO is a new swarm intelligence optimization algorithm proposed by Heidari et al.[9] in 2019 that mimics the way Harris eagles find and chase prey in nature, including global exploration, local besiege and pounce behaviour. However, similar to the other SI algorithms, the HHO still has some limitations, such as the multiplicity of solutions generated by a randomized policy that is finite in the initialization phase. Moreover, because global exploration is only performed in the first half of the iteration, it is difficult to balance the global exploration and local exploitation capacities by using the escaping energy of prey, so the algorithm may converge slowly, has low solution accuracy and prematurely falls into a local optimal solution. In this paper, to conquer the limitations of HHO, a chaotic sequence chaotic sequence recombination mechanism (CSRM) strategy is proposed, which enhances the distribution of the initialized solutions in the search space, and accelerates the convergence rate of HHO, and the generalized opposition-based learning recombination mechanism (OBLRM) is proposed, which can have the opportunity to carry out global search in the later period of iteration to jump out of the local optimum and improve the accuracy of the solution.

---

## [Decision Letter · Decision Letter 1]

4 Aug 2022

PONE-D-22-04874R1An Improved Harris Hawks Optimization Algorithm based on Chaotic Sequence and Opposite Elite Learning MechanismPLOS ONE

Dear Dr. Jia,

Thank you for submitting your manuscript to PLOS ONE. After careful consideration, we feel that it has merit but does not fully meet PLOS ONE’s publication criteria as it currently stands. Therefore, we invite you to submit a revised version of the manuscript that addresses the points raised during the review process.

We look forward to receiving your revised manuscript.

Kind regards,

Pijush Samui

Academic Editor

PLOS ONE

Reviewers' comments:

Reviewer's Responses to Questions

**Comments to the Author**

1. If the authors have adequately addressed your comments raised in a previous round of review and you feel that this manuscript is now acceptable for publication, you may indicate that here to bypass the “Comments to the Author” section, enter your conflict of interest statement in the “Confidential to Editor” section, and submit your "Accept" recommendation.

Reviewer #1: (No Response)

2. Is the manuscript technically sound, and do the data support the conclusions?

Reviewer #1: (No Response)

3. Has the statistical analysis been performed appropriately and rigorously? 

Reviewer #1: (No Response)

4. Have the authors made all data underlying the findings in their manuscript fully available?

Reviewer #1: (No Response)

5. Is the manuscript presented in an intelligible fashion and written in standard English?

Reviewer #1: (No Response)

6. Review Comments to the Author

Reviewer #1: 1. The authors must add section number to get a clear view.

2. In the Introduction section, there must be point-wise contribution and the structure of the paper must be given in the last paragraph of the Introduction.

3. Please note that the caption style of those figures is wrong in which there are multi-figures.

4. More discussions on results are required.

5. What are the practical use-cases of the proposed scheme?

6. Improve the English language.

7. In the Reference section, C, J, etc. are mentioned with each reference. Please delete them. Replace reference no. 3 with https://doi.org/10.1049/cit2.12040. Also, include 10.4018/IJDWM.2020070104.

7. PLOS authors have the option to publish the peer review history of their article (what does this mean?). If published, this will include your full peer review and any attached files.

Reviewer #1: No

---

## [Author Response · Author response to Decision Letter 1]

14 Aug 2022

Dear reviewers, 

These are my responses to each question. Thank you for your advice in your busy schedule, I thought about each question carefully and gave an explanation. Due to my limited research skills, there may be some incorrect answers, please criticize and correct me. 

Thanks again! Best wishes to you!

Response to Reviewer #1: 

1. The authors must add section number to get a clear view.

Reply: I have modified it.

2. In the Introduction section, there must be point-wise contribution and the structure of the paper must be given in the last paragraph of the Introduction.

Reply: I have modified it. The contributions of this paper are as follows: (i) a chaotic sequence chaotic sequence recombination mechanism (CSRM) strategy is proposed, which enhances the distribution of the initialized solutions in the search space, and accelerates the convergence rate of HHO;(ii) the generalized opposition-based learning recombination mechanism (OBLRM) is proposed, which can have the opportunity to carry out global search in the later period of iteration to jump out of the local optimum and improve the accuracy of the solution. The rest of the paper is organized as follows. Section 2 gives a detailed overview of the HHO algorithm. The proposed method is introduced in detail in Section 3. In Section 4, the experimental results are analysed. Finally, the conclusions are presented in Section 5.

3. Please note that the caption style of those figures is wrong in which there are multi-figures.

Reply: I have modified it.

4. More discussions on results are required.

Reply: I have modified it. In this paper, we propose a new HHO variant algorithm based on chaotic mapping and an opposite elite learning mechanisms. The chaotic sequence recombination mechanism is applied at the initial population stage to improve the diversity of the population and enhance the exploration ability of the HHO algorithm. The opposite elite learning recombination mechanism is used at the last stage of each iteration, to effectively maintain the optimal individual and enhance the exploitation ability. On the other hand, this method can overcome the shortcoming at the late iteration in the HHO algorithm: inability to perform global search. In addition, the exploitation and exploration capabilities of the HHO algorithm are balanced. The performance of the HHO-CS-OELM algorithm is verified by comparison with 14 optimization algorithms on 23 benchmark functions and an engineering problem which aims to optimize the weights and thresholds of the BP neural network for indoor positioning. Experimental analyses revealed that the proposed algorithm can obtain optimal values for 11 functions out of 23 functions compared to other state-of-the-art SI algorithms, while it outperforms the standard HHO algorithm on all of the 23 functions, so the HHO-CS-OELM algorithm offers competitive results compared to other state-of-the-art SI algorithms, and verified that the HHO-CS-OELM algorithm has better performance than others. The improved BP neural network optimized by the HHO-CS-OELM algorithm reduced the indoor average positioning error from 7.76 cm to 3.81 cm which is obtained by BP neural network, it also revealed that the indoor positioning accuracy obtained by using the BP neural network with the proposed algorithm is significantly improved, which further verifies the superiority of the HHO-CS-OELM algorithm.

To further improve the performance of the HHO-CS-OELM algorithm, outcomes of other benchmark functions still need to be enhanced, and the exploitation and exploration capabilities of the HHO-CS-OELM algorithm need to be investigated further. Moreover, the HHO-CS-OELM algorithm should be used to solve the system optimal parameters, optimal parameters of neural network model, and so on.

5. What are the practical use-cases of the proposed scheme?

Reply: The proposed scheme in this paper mainly improved the performance of HHO algorithm, and then improved the solution accuracy of HHO algorithm in the engineering application field. The practical use-cases included the solution of indoor positioning, the solution of system optimal parameters, and so on.

6. Improve the English language.

Reply: This comment is crucial to improve the quality of my paper. Thank you very much for your comment, I also try to improve my English language to the best of my ability, but English language is not my mother tongue but my second language, so the English language is still to be further improved. I have improved the English language by AJE, the verification code is E65D-842F-B583-0487-1D9C, the URL is https://secure.aje.com/en/certificate/verify. 

7. In the Reference section, C, J, etc. are mentioned with each reference. Please delete them. Replace reference no. 3 with https://doi.org/10.1049/cit2.12040. Also, include 10.4018/IJDWM.2020070104.

Reply: I have delete them, and I replace reference no. 3 with https://doi.org/10.1049/cit2.12040 and 10.4018/IJDWM.2020070104.

---

## [Editor Report · Decision Letter 2]

30 Jan 2023

An Improved Harris Hawks Optimization Algorithm based on Chaotic Sequence and Opposite Elite Learning Mechanism

PONE-D-22-04874R2

Dear Dr. Jia,

We’re pleased to inform you that your manuscript has been judged scientifically suitable for publication and will be formally accepted for publication once it meets all outstanding technical requirements.

Kind regards,

Pijush Samui

Academic Editor

PLOS ONE
---

## [Editor Report · Acceptance letter]

10 Feb 2023

PONE-D-22-04874R2 

An Improved Harris Hawks Optimization Algorithm based on Chaotic Sequence and Opposite Elite Learning Mechanism 

Dear Dr. Jia:

I'm pleased to inform you that your manuscript has been deemed suitable for publication in PLOS ONE. Congratulations! Your manuscript is now with our production department. 

Kind regards, 

on behalf of

Dr. Pijush Samui 

Academic Editor

PLOS ONE